# IDENTIFYING OPTIMAL OUTPUT SETS FOR DIFFERENTIAL PRIVACY AUDITING

## ABSTRACT

Differential privacy limits an algorithm's privacy loss, defined as the maximum influence *any* individual data record can have on the probability of observing *any* possible output. Privacy auditing identifies the worst-case input datasets and output event sets that empirically maximize privacy loss, providing statistical lower bounds to evaluate the tightness of an algorithm's differential privacy guarantees. However, current auditing methods often depend on heuristic or arbitrary selections of output event sets, leading to weak lower bounds. We address this critical gap by introducing a novel framework to compute the *optimal output event set* that maximizes the privacy loss lower bound in auditing. Our algorithm efficiently computes this optimal set when closed-form output distributions are available and approximates it using empirical samples when they are not. Through extensive experiments on both synthetic and real-world datasets, we demonstrate that our method consistently tightens privacy lower bounds for auditing differential privacy mechanisms and black-box DP-SGD training. Our approach outperforms existing auditing techniques, providing a more accurate analysis of differentially-private algorithms.

## 1 INTRODUCTION

Differential privacy (DP) (Dwork et al., 2014) is regarded as the gold standard for quantitatively assessing an algorithm's privacy guarantee. It ensures that the presence or absence of any record will not be heavily revealed from the outcome of the mechanism. Formally, consider a randomized algorithm $M : \mathcal{D} \to \Theta$. If for any neighbouring datasets $D, D' \in \mathcal{D}$ that differ by only one entry, i.e., $D' = D \cup \{x\}$ or $D' = D \setminus \{x\}$, and for any measurable (output event) sets $\mathcal{O}$, it satisfies that

$$\mathbb{P}[M(D) \in \mathcal{O}] \leq e^\epsilon \cdot \mathbb{P}[M(D') \in \mathcal{O}] + \delta,$$

then the algorithm $M$ is said to satisfy $(\epsilon, \delta)$-DP. Differential privacy mathematically certifies that the information leakage of a randomized algorithm is bounded (by $\varepsilon$ and $\delta$) even under the *worst-case* datasets $D, D'$ and the *worst-case* output event set $\mathcal{O}$.

Computing the exact differential privacy loss $\varepsilon$ and $\delta$ for an algorithm requires tracking its output distribution over all possible input datasets and output event sets, which is challenging for complex algorithm (e.g., iterative learning algorithms with a large number of non-linear updates). To circumvent this challenge, existing privacy analysis for machine learning algorithms (Abadi et al., 2016; Kairouz et al., 2015; Mironov, 2017) often make simplifying assumptions that the adversary observes not only the final output, but also other information (such as intermediate updates) in a learning algorithm, and prove *upper bounds* for $\varepsilon$ and $\delta$ instead. Understanding whether the DP upper bound is correct and tight is critical, since underestimating privacy upper bounds leads to privacy leaks (Tramer et al., 2022), while any underestimation results in unnecessarily large modifications to the learning algorithms which decreases the utility of the trained model. To measure the tightness of the DP upper bound, it is crucial to identify the worst-case datasets and output event sets where the privacy loss is maximized. The success of inference attack under such worst-case scenarios then provides a statistical lower bound on the exact differential privacy loss of the algorithm. If the lower bound is close to the DP upper bound, then the privacy analysis is tight. This process is known as *privacy auditing* (Jagielski et al., 2020; Nasr et al., 2021; Zanella-Beguelin et al., 2023; Nasr et al., 2023; Steinke et al., 2023).

Existing privacy auditing literature largely focuses on designing the worst-case input datasets $D, D'$ via poisoning attacks (Jagielski et al., 2020; Nasr et al., 2021) or via searching for the data with the highest risks against inference attacks in the training dataset via computationally expensive inference attacks (Ye et al., 2022; Zanella-Beguelin et al., 2023; Aerni et al., 2024). By contrast, the question of how to choose the worst-case output event set $\mathcal{O}$ to enable a better auditing lower bound is largely underexplored. This is due to several reasons. **(1)** Standard learning algorithms consist of a large number of iterative non-linear updates, which makes it **challenging to track the complex distribution of the final output** in a *closed-form* manner. Consequently, it is *infeasible* to perform *exact* optimization of output event set for complex iterative learning algorithms. **(2)** If we relax ourselves to *approximate* optimization of output set, prior works (Bichsel et al., 2021; Lu et al., 2022) have proposed several approaches based on approximating the output distributions via their empirical Monte Carlo samples. However, such approximations require a large number of Monte Carlo samples to be accurate, which requires computationally expensive (re)runs of the training algorithm. Alternatively, given a small finite number of Monte Carlo samples, such methods suffer from non-negligible approximation error, which in turn affects the auditing performance. Thus, it is *crucial* to *take the approximation error into account when formulating the objective of maximal privacy loss in privacy auditing*. However, to the best of our knowledge, there is **no existing consensus regarding the optimal way to incorporate the approximation error into the outpupt set selection objective for privacy auditing**. **(3)** Prior works (Steinke et al., 2023; Bichsel et al., 2021; Lu et al., 2022) have empirically observed that the *effect of output event set choice on the privacy auditing power is distribution-dependent*. For example, in certain mechanisms like randomized response, the choice of output event set does not significantly affect the audited lower bound ((Steinke et al., 2023, Figure 10)). In contrast, for some other mechanisms (such as the Laplace mechanism), the choice of output event set substantially affects the audited lower bound Bichsel et al. (2021); Lu et al. (2022). Despite these observations, **a comprehensive understanding of whether and when optimizing the output event set yields a gain remains elusive** (i.e., how does the amount of gain change for different mechanisms).

In this paper, we aim to address the question of optimal output set selection for privacy auditing. Specifically, we ask the following research questions.

1. What is the optimal output event set for auditing differentially private mechanisms? How does the optimal output event set change for different mechanisms?

2. How much gain for privacy auditing can we achieve by optimizing the output set? How does the amount of gain depend on the mechanism and the number of auditing examples?

3. How to compute the optimal output event set when only given empirical output samples, rather than closed-form output distributions? Does this enable tighter auditing lower bounds for practical DP learning mechanisms, such as black-box DP-SGD?

***Summary of Results***. Our results for addressing the above research questions are as follows.

- We proposed a novel framework for choosing the optimal output set in privacy auditing, that incorporates the error of sampling auditing examples from output distributions. Specifically, our optimization objective (4) captures the expectation of advantage-based auditing lower bound (Steinke et al., 2023; Jayaraman & Evans, 2019) over random sampling.

- We prove that our proposed optimization problem incurs closed-form solutions, that consist of output sets constructed by thresholding the likelihood ratio between member and non-member score distributions in privacy auditing (Definition 4.2). Importantly, we prove (Theorem 4.3) that our constructed output set enables the *maximum auditing objective* among all possible choices of output event set, i.e., is optimal.

- We numerically validate that for auditing DP mechanisms with closed-form output distributions, using the optimal output set *consistently* enables tighter lower bounds than prior methods. Our improvement is especially significant under a small number of auditing examples (Figure 2), or when the output distribution is asymmetric or multi-modal (Figure 1).

- We also extend our framework to algorithms *without* closed-form output distributions (Section 4.3), and approximately compute the optimal output set by thresholding the KDE estimation of likelihood ratio from empirical output samples. We experimentally demonstrate

(Section 6) that our algorithm is effective and enables tighter privacy auditing of black-box DP SGD training on synthetic (canary) and real-world (CIFAR-10) datasets.

## 2 RELATED WORKS

***Score-based membership inference attacks and connections to output set selection***. Membership inference attacks (MIAs) aim to infer whether a given data record was used for training a given target model. Intuitively, a higher performance of MIA indicates higher privacy risks, and thus MIAs serve as crucial tools for auditing DP mechanisms (Shokri et al., 2017). MIAs in the literature generally follow a fixed template: compute a membership inference attack score (such as loss (Sablayrolles et al., 2019; Yeom et al., 2018; Ye et al., 2022), confidence (Salem et al., 2018), entropy (Song & Mittal, 2021), loss trajectory (Liu et al., 2022) and likelihood ratio (Carlini et al., 2022)) on the target model and target data, and then perform attack via thresholding the MIA score. That is, predict member if the MIA score is above the threshold, and predict non-member if the MIA score is below the threshold. To this end, performing score-based membership inference attacks (on the output distributions in privacy auditing) is *equivalent* to heuristically selecting the output event set to be the interval $(Z, \infty)$ by a threshold $Z$, as utilized in the auditing experiment in Jagielski et al. (2020). This strategy, as we will show in Section 4, 5 and 6, is strictly suboptimal for certain DP mechanisms compared to our proposed optimization-based output event set selection.

***Optimization of output event set in privacy auditing***. Bichsel et al. (2021) optimizes the posterior over output event set, while ensuring that the probability of output event set is larger than a heuristically selected threshold (e.g., 0.05). Lu et al. (2022) further trains a ML model to learn the posterior of training data given trained model, and selects the output set that maximizes the posterior. Steinke et al. (2023) considers a series of output event sets constructed by a fixed strategy. Their constructed output event set takes the form of two intervals $(-\infty, c_-)$ and $(c_+, \infty)$, where $c_-$ and $c_+$ are real-valued thresholds for the MIA scores. They then search over the entire output domain (i.e., the set of values for discrete Monte Carlo samples) to identify a choice of $c_-$ and $c_+$ that enables maximum empirical auditing performance. However, prior works do not reach a consensus regarding the optimal output set for privacy auditing. By contrast, our work proposes a novel objective (4) that captures the expectation of auditing lower bound and serves as the key ingredient for us to prove an optimality guarantee for selecting output set in privacy auditing given finite auditing samples.

***Evaluating worst-case success of membership inference attacks***. A connected direction of research is on measuring privacy risk of machine learning algorithms by the success of membership inference attack over a small set of (worst-case) data samples, rather than the entire dataset. To this end, Carlini et al. (2022) propose to evaluate the true positive rate (TPR) at a low false positive rate (FPR) as an indicator of privacy loss as worst-case data records. This is equivalent to evaluating MIA only on a selected output set at the bottom tail of the MIA score distributions on all training and test data (as this constrains the FPR to be small), rather than the whole output domain.

## 3 PROBLEM STATEMENT

Let $\mathcal{T} : D \mapsto \theta$ be an $(\varepsilon, \delta)$-differentially private algorithm, that maps an input dataset $D$ to a trained model with parameters $\theta$. Let $Score(\theta, x)$ be a MIA score function, that maps the algorithm's output model parameters $\theta$ and any data record $x$ to a real-valued score $o$. An auditing experiment $\mathcal{E}$ takes in an randomized algorithm $\mathcal{T}$, a confidence tolerance $\beta$, and an approximate differential privacy tolerance $\delta$ as inputs, and returns a high-confidence statistical lower bound $\hat{\varepsilon} := \mathcal{E}(\mathcal{T})$ for the ground-truth differential privacy parameter $\varepsilon$ that satisfies

$$\Pr[\hat{\varepsilon} := \mathcal{E}(\mathcal{T}) < \varepsilon] \geq 1 - \beta \quad (1)$$

where the probability is over randomness of the auditing experiment $\mathcal{E}$ and the training algorithm $\mathcal{T}$. Generally speaking, existing auditing experiments (Jagielski et al., 2020; Nasr et al., 2021; 2023; Zanella-Beguelin et al., 2023; Bichsel et al., 2021; Lu et al., 2022) (see Appendix A and Table 1 for more details) can be decomposed into the following three components.

1. **Compute a set of member scores $S_+$ and non-member scores $S_-$.** This involves sampling the input dataset, training output models, and computing MIA scores on the trained model(s) and its member and non-member data record(s) respectively.

2. **Subselect the scores by a output set $\mathcal{O} \subseteq \mathbb{R}$.** Intuitively, this is to increase the distinguishability between subselected member and non-member scores (that lie in $\mathcal{O}$), so as to enable a higher auditing lower bound.

3. **Design auditing function $L$ such that $L(S_+, S_-, \mathcal{O}; \delta, \beta) \leq \varepsilon$ with high probability.** Typically, $L$ is a function of the performance of membership inference in auditing. [1] To prove $L$ satisfies (1), prior works generally rely on the constraints of $(\varepsilon, \delta)$-DP on the power of binary membership hypothesis test (Wasserman & Zhou, 2010; Kairouz et al., 2015).

**Our objective** We aim to design a general framework for computing the optimal output set, so as to enable the highest possible lower bound $\hat{\varepsilon}$ in privacy auditing. Formally speaking, for any fixed distribution of member scores $S_+$ and non-member scores $S_-$ (specified by the data sampling and training protocols in the auditing experiment), any approximate DP tolerance $\delta$, any confidence tolerance $\beta$, and any auditing function $L$, we aim to solve the below optimization problem.

$$\arg\max_{\mathcal{O}} \hat{\varepsilon} = L(S_+, S_-, \mathcal{O}; \delta, \beta) \tag{2}$$

By fixing $S_+, S_-, \delta, \beta$, and $L$, we are essentially fixing the first and third components of the auditing experiment. For simplicity of presentation, and for fair comparison with prior works (Bichsel et al., 2021; Lu et al., 2022) that solely focus on pure DP settings, in the paper, we only optimize and report auditing lower bounds for $\delta = 0$ and $\beta = 0.05$. To compute the member and non-member scores $S_+, S_-$, we focus on two representative ways in the literature (Jagielski et al., 2020; Steinke et al., 2023) as summarized by Table 1. For auditing function, we will use generalized variants (Proposition 4.1) of the advantage-based auditing function in (Steinke et al., 2023, Theorem 5.2). [2] However, we emphasize that we aim to design a general framework for optimizing the output set that can be combined with any member and non-member scores and auditing functions in the literature.

## 4 COMPUTING THE OPTIMAL OUTPUT SET FOR PRIVACY AUDITING

In this section, we introduce our optimization objective for selecting the output set to enable higher advantage-based privacy auditing lower bound. We focus on the setting where the member and non-member scores $S_+$ and $S_-$ are i.i.d. samples from two output distributions $p$ and $q$ respectively. This is indeed the case for prototypical auditing experiments (Jagielski et al., 2020; Nasr et al., 2021) – see Appendix A and Table 1 for details. We then analyze the structure of the optimal output set and propose an algorithm to approximate the optimal output set from empirical samples.

### 4.1 OPTIMIZATION OBJECTIVE FOR OUTPUT SET SELECTION

Given black-box access to i.i.d. samples from the output distributions of the DP mechanism, we first prove a new auditing function that has explicit dependence on the choice of output set $\mathcal{O}$ as follows.

**Proposition 4.1** (Our Output-set-dependent Auditing Function). *Let $p$ and $q$ be two probability distributions over $\mathbb{R}$ that satisfies $e^{-\varepsilon} \leq \frac{p(o)}{q(o)} \leq e^{\varepsilon}$ for any $o \in \mathbb{R}$. Let $\beta = 0.05$ be the confidence tolerance. Let $\mathcal{O} \subset \mathbb{R}$ be a fixed output set. Let $S_+$ be $m$ i.i.d. samples from $p$, and let $S_-$ be $m$ i.i.d. samples from $q$. Then $\hat{\varepsilon}(S_+, S_-, \mathcal{O}; p, q)$ defined as follows is a valid auditing function.*

$$\hat{\varepsilon}(S_+, S_-, \mathcal{O}; p, q) = \phi\left(\frac{\int_{o \in \mathcal{O}} \max\{p(o), q(o)\} do}{p(\mathcal{O}) + q(\mathcal{O})} - \sqrt{\frac{2 \cdot \ln(2/\beta)}{|S_+ \cap \mathcal{O}| + |S_- \cap \mathcal{O}|}}\right) \tag{3}$$

*where $\phi(y) = \log\left(\frac{y}{1-y}\right)$ is the logit function. That is, $\Pr[\hat{\varepsilon}(S_+, S_-, \mathcal{O}; p, q) \leq \varepsilon] \geq 1 - \beta$.*

*Proof.* The proof is in Appendix B.3. This generalizes prior auditing functions Yeom et al. (2018); Steinke et al. (2023) from specific output sets to any $\mathcal{O}$, and enjoys the benefit of explicit dependence on the output set – see Remark B.4 and B.5 for details. $\square$

---

[1] Other auditing functions include the ones based on attribute inference Guo et al. (2023), reconstruction Guo et al. (2022); Balle et al. (2022), and empirical divergence estimation Domingo-Enrich & Mroueh (2022); Kong et al. (2024). It is interesting to invetigate the effect of output set optimization for them as future works.

[2] That is, the design of $L$ is based on bounding the advantage of membership inference with the DP guarantee $(\varepsilon, \delta)$. Extending our framework to other performance metrics of MIA is an interesting future work.

By taking expectation of the denominator $|S_+ \cap \mathcal{O}| + |S_- \cap \mathcal{O}|$ in (3) across random sampling of $S_+$ and $S_-$, we obtain an objective for selecting the optimal output set $\mathcal{O}$ in privacy auditing.

$$\arg\max_{\mathcal{O}} \hat{\varepsilon}(\mathcal{O}; p, q) := \phi \left( \underbrace{\frac{\int_{o \in \mathcal{O}} \max\{p(o), q(o)\} do}{p(\mathcal{O}) + q(\mathcal{O})}}_{\text{inference accuracy}} - \underbrace{\sqrt{\frac{2 \cdot \ln(1/\beta)}{m \cdot (p(\mathcal{O}) + q(\mathcal{O}))}}}_{\text{sampling error}} \right) \quad (4)$$

where $\phi(y) = \log\left(\frac{y}{1-y}\right)$. Note that the objective (4) is specified by distributions $p$ and $q$, indicating that the optimal $\mathcal{O}$ is mechanism-dependent. We now analyze the structure of the optimal output set.

## 4.2 Identifying Optimal Output Set for Auditing

We first define a series of output set $\mathcal{O}_\tau$ by thresholding the likelihood ratio between distributions.

**Definition 4.2** ($\tau$-log-likelihood-ratio-set). Let $p$ and $q$ be two continuous distributions over $\mathbb{R}$, indicating member and non-member score distributions in the auditing experiment respectively. For each $\tau > 0$, we define the $\tau$-log-likelihood-ratio-set for $p$ and $q$ as the following set $\mathcal{O}_\tau$.

$$\mathcal{O}_\tau = \left\{ x \in \mathbb{R} : \left| \ln\left(\frac{p(x)}{q(x)}\right) \right| \geq \tau \right\} \quad (5)$$

We now prove the optimality guarantee for $\mathcal{O}_\tau$, i.e., $\mathcal{O}_\tau$ enables the largest lower bound among all possible output sets $S$ that satisfy the same size constraint $p(\mathcal{O}) + q(\mathcal{O}) = p(\mathcal{O}_\tau) + q(\mathcal{O}_\tau)$.

**Theorem 4.3.** *Let $p$ and $q$ be two continuous distributions over $\mathbb{R}$. Let $\hat{\varepsilon}(\cdot; p, q)$ be our auditing objective* (4). *Given any feasible output set $\mathcal{O}$, there exists $\tau \in \mathbb{R}$ such that $p(\mathcal{O}_\tau) + q(\mathcal{O}_\tau) = p(\mathcal{O}) + q(\mathcal{O})$, where $\mathcal{O}_\tau$ be the $\tau$-log-likelihood-ratio-set* (5) *for $p$ and $q$. Further, it satisfies that*

$$\hat{\varepsilon}(\mathcal{O}_\tau; p, q) = \max_{\mathcal{O} \subseteq \mathbb{R}: p(\mathcal{O}) + q(\mathcal{O}) = p(\mathcal{O}_\tau) + q(\mathcal{O}_\tau)} \hat{\varepsilon}(\mathcal{O}; p, q) \quad (6)$$

The proof technique (Appendix B.2) is similar to the Neyman-Pearson Lemma (Neyman & Pearson, 1933) (Remark B.6). Theorem 4.3 proves that for any output set $\mathcal{O}$, there exists a $\tau$-log-likelihood-ratio-set $\mathcal{O}_\tau$ that satisfies $p(\mathcal{O}_\tau) + q(\mathcal{O}_\tau) = p(\mathcal{O}) + q(\mathcal{O})$ such that $\mathcal{O}_\tau$ enables higher auditing lower bound objective (4) than $\mathcal{O}$. Thus the family of $\{\mathcal{O}_\tau\}_{\tau \geq 0}$ contains the optimal output set.

## 4.3 Approximating the Optimal Output Set from Empirical Samples

In practice, we are only given empirical samples from the output distributions $p$ and $q$, rather than the closed-form densities. We now propose a method to approximate the optimal output set $\mathcal{O}_\tau$ from empirical samples. The idea is to estimate the probability densities of distributions $p$ and $q$ from their empirical samples (e.g., via kernel density estimation), and then compute the likelihood ratio function and its level set as optimal output set. See Algorithm 1 for the pseudocode.

---

**Algorithm 1** Approximating the Optimal Optimal Output Set from Empirical Samples

---

**Require:** $m_+$ samples $S_+$ and $m_-$ samples $S_-$ from unknown distributions $p, q$ respectively, with $m_+ + m_- = 2m$. DP tolerance $\delta$. Confidence $\beta$. Auditing function $L(S_+, S_-, \mathcal{O}; \delta, \beta)$.
1: Kernel density estimation (KDE) from samples: $\hat{p} \leftarrow KDE(S_+), \hat{q} \leftarrow KDE(S_-)$
2: Sort the empirical samples: $\tilde{o}_0 = -\infty; \tilde{o}_1 \leq \cdots \leq \tilde{o}_{2m} \leftarrow \text{sort}(S_+ \cup S_-); \tilde{o}_{2m+1} = +\infty$
3: Estimate absolute log likelihood ratio: $\{\tau_0, \cdots, \tau_{2m}\} := \left\{ \left| \log \frac{\int_{\tilde{o}_i}^{\tilde{o}_{i+1}} \hat{p}(o) do}{\int_{\tilde{o}_i}^{\tilde{o}_{i+1}} \hat{q}(o) do} \right| \right\}_{i=0}^{2m}$
4: **for** each level $\tau \in \{\tau_0, \cdots, \tau_{2m}\}$ **do**
5:     approximate the $\tau$-log-likelihood-ratio set: $\mathcal{O}_\tau = \cup_{i=0: \tau_i \geq \tau}^{2m} (\tilde{o}_i, \tilde{o}_{i+1})$.
6: Search the optimal level $\hat{\tau} := \arg\max_{\tau \in \{\tau_0, \cdots, \tau_{2m}\}} L(S_+, S_-, \mathcal{O}_\tau; \delta, \beta)$.
7: **return** output set $\mathcal{O}_{\hat{\tau}}$.

---

**On reducing the instability of KDE and likelihood ratio estimation** Motivated by Bichsel et al. (2021), we only perform likelihood ratio estimation on the regions where the estimated densities are non-negligible (with probability density function $\hat{p}(o)$ and $\hat{q}(o)$ that is larger than 0.01). This is because the likelihood ratio estimation involves division over probability density, and thus the error can be arbitrarily large when the estimated densities are small.

**On searching for the optimal leve** $\hat{\tau}$    Theorem 4.3 proves that the family of $\tau$-log-likelihood-ratio-set contains the optimal output set. To choose one single output set over the family of $\mathcal{O}_\tau$, in Line 6 of Algorithm 1, we evaluate the auditing performance of $\mathcal{O}_\tau$ for different values $\tau$ and return *one* output set with the best performance. This is a one-dimensional optimization problem over $\tau \in \mathbb{R}$, which is significantly easier and incurs significantly less computation cost than the original output set optimization problem over all possible output sets $\mathcal{O} \subseteq \mathbb{R}$. Although this search for optimal threshold requires additional output samples, it is a common practice in the literature (Bichsel et al., 2021; Lu et al., 2022; Zanella-Beguelin et al., 2023) to boost the auditing performance.

## 5    NUMERICAL EXPERIMENTS: ONE-EPOCH OF SHUFFLED NOISY SGD

In this section, we perform numerical experiments on two fundamental differentially private mechanisms: **Lapalace mechanism** (which is pure DP), and **Gaussian mechanism** (which is approximate DP), that are *building blocks for the iterative update in the celebrated DP-SGD learning algorithm* (Abadi et al., 2016). We consider *composition of both mechanisms* for *one shuffled epoch* of noisy stochastic gradient descent udpates (Bassily et al., 2014), under simple loss functions construction. We then investigate whether our output set optimization method enables tighter privacy auditing lower bound than prior works (Jagielski et al., 2020; Bichsel et al., 2021; Lu et al., 2022; Steinke et al., 2023), and how various factors affect the amount of gain.

**Experiment setting**    Consider a simplified setting of learning on dataset $D = (x_1, x_2)$ with two records $x_1, x_2 \in \mathbb{R}$, where the loss function is $\ell_1(x, \theta) = \langle \theta, x \rangle$ for the first record, and is $\ell_2(x, \theta) = \frac{1}{2}(\theta - x)^2$ for the second record. Assuming different loss function for different records of the dataset is realistic, for example, in a multi-task learning setting, where the first record is used for learning task one (e.g., classify between cats and dogs) and the second record is used for learning task two (e.g., classify between sketch and photo). In terms of learning algorithm, we consider running one epoch of shuffled noisy stochastic gradient descent algorithm with the following updates:

$$\theta_1 = \theta_0 - \eta \left( \nabla \ell_1(\theta_0, x_{s_1}) + Z_1 \right) \quad \text{where } Z_1 \sim \text{Lap}(0, b_1) \text{ or } \mathcal{N}(0, \sigma_1^2)$$

$$\theta_2 = \theta_1 - \eta \left( \nabla \ell_2(\theta_1, x_{s_2}) + Z_2 \right) \quad \text{where } Z_2 \sim \text{Lap}(0, b_0.5) \text{ or } \mathcal{N}(0, \sigma_2^2) \tag{7}$$

where $\theta_0$ is the initialization parameters of the model (assumed to be zero for simplicity), $\eta$ is the learning rate, $b_1, b_0.5, \sigma_1$ and $\sigma_2$ are the noise scales, and $s \xleftarrow{\text{uniform}} \{(1, 2), (2, 1)\}$ is a randomly shuffled dataset order. Under the assumption of bounded data domain, the gradient of both loss functions $\ell_1(\theta, x)$ and $\ell_2(\theta, x)$ have finite sensitivity, thus the algorithm satisfies differential privacy by standard DP guarantees of Laplace mechanism (Dwork et al., 2006, Theorem 1) and Gaussian mechanism (Balle & Wang, 2018, Theorem 8). (See Appendix C for the details.) Conditioned on a fixed order $s$, the final output $\theta_2$ of this mechanism follows a closed-form distribution as follows.

$$\theta | s \stackrel{d}{=} \theta_0 - \eta \cdot (1 - \eta) \cdot x_{s_1} + \eta \cdot x_{s_2} + Z \quad \text{where } Z = -\eta(1 - \eta)Z_1 - \eta Z_2 \tag{8}$$

For simplicity, assume that $\theta_0 = 0$ and $\eta = \frac{1}{2}$ (note that both terms do not affect the DP guarantee of the mechanism in Proposition C.1). By averaging over the two possible orders $s$, we could obtain the closed-form output distributions $p(\theta)$ and $q(\theta)$ for model $\theta$ trained on dataset $D = (x_1, x_2)$ and $D' = (x_1', x_2')$ respectively. For example, for Gaussian mechanism, when $Z_1 \sim \mathcal{N}(0, \sigma^2)$ and $Z_2 \sim \mathcal{N}(0, \sigma^2)$, we have that $p(\theta)$ and $q(\theta)$ follows the below mixture distributions.

$$p(\theta) = \frac{1}{2} \cdot \mathcal{N} \left( -\frac{1}{4}x_1 + \frac{1}{2}x_2, \frac{5}{16}\sigma^2 \right) + \frac{1}{2} \cdot \mathcal{N} \left( -\frac{1}{4}x_2 + \frac{1}{2}x_1, \frac{5}{16}\sigma^2 \right) \tag{9}$$

$$q(\theta) = \frac{1}{2} \cdot \mathcal{N} \left( -\frac{1}{4}x_1' + \frac{1}{2}x_2', \frac{5}{16}\sigma^2 \right) + \frac{1}{2} \cdot \mathcal{N} \left( -\frac{1}{4}x_2' + \frac{1}{2}x_1', \frac{5}{16}\sigma^2 \right) \tag{10}$$

Similarly, for Laplace mechanism, when $Z_1 = 0 \sim \text{Lap}(0, 0)$ and $Z_2 \sim \text{Lap}(0, b)$, we have that $p(\theta)$ and $q(\theta)$ follows the below mixture distributions.

$$p(\theta) = \frac{1}{2} \cdot \text{Lap} \left( -\frac{1}{4}x_1 + \frac{1}{2}x_2, \frac{1}{2}b \right) + \frac{1}{2} \cdot \text{Lap} \left( -\frac{1}{4}x_2 + \frac{1}{2}x_1, \frac{1}{2}b \right) \tag{11}$$

$$q(\theta) = \frac{1}{2} \cdot \text{Lap} \left( -\frac{1}{4}x_1' + \frac{1}{2}x_2', \frac{1}{2}b \right) + \frac{1}{2} \cdot \text{Lap} \left( -\frac{1}{4}x_2' + \frac{1}{2}x_1', \frac{1}{2}b \right) \tag{12}$$

Equation (9), 10, 11 and 12 are the forms of output distribution $p$ and $q$ that we will use for auditing one-epoch shuffled noisy SGD in this section. That is, to compute member and non-member scores $S_+$ and $S_-$, we perform Monte Carlo sampling from the output distributions of the mechanism. This is equivalent to the auditing experiment via retraining in Jagielski et al. (2020).

**Comparison Baselines** For our method, we approximate the optimal output set by applying Algorithm 1 on top of black-box Monte Carlo samples from the output distributions( (9), 10, 11 and 12) and auditing function add ref for i.i.d. samples given by prior works . To validate the optimality guarantee (Theorem 4.3), we compare with the following methods.

1. **Whole domain:** A naive strategy is to choose $\mathcal{O}$ as $\mathbb{R}$, i.e., the whole score domain.

2. **Top (bottom) score heuristic:** (Steinke et al., 2023, Algorithm 1) uses a strategy of 'abstaining' by only performing MIAs on the top $k^+$ and bottom $k_-$ scores in the auditing experiment, which is equivalent to the output set spanned by top $\alpha_+$-percentile and bottom $\alpha_-$-percentile of the mixture of member and non-member scores. This selection intuitively restricts the guesses to scores that are most likely to be members or non-members, thus boosting the attack accuracy and the auditing lower bound. Experimentally, (Steinke et al., 2023, Figure 11) observes that the choice of $k^+$ and $k_-$ has a significant impact on the auditing lower bound, indicating that this heuristic outperforms the naive baseline of using the whole score domain. In our experiments, we follow (Steinke et al., 2023) and brute-force search over all possible values to find the best $k_+$ and $k_-$, and then report the auditing performance of the best $k_+$ and $k_-$ in fresh evaluation trials.

3. **DP-sniper (Bichsel et al., 2021) and Lu et al. (2022):** Bichsel et al. (2021) propose to compute output set by thresholding the *posterior* probability of training dataset conditional on mechanism output. Lu et al. (2022) further makes the method more flexible by searching for the optimal threshold that enables the largest auditing lower bound. In experiments, we use the implementation [3] released by Bichsel et al. (2021) and additionally follow Lu et al. (2022) to search for the optimal threshold for fair comparison.

For all methods, we directly use the one-dimensional output of the DP mechanisms as the MIA score and subsequently perform the optimal advantage-based MIA with rejection region $A = \{o : \hat{p}(o) \geq \hat{q}(o)\}$ (benefited from the KDE estimators of the output distributions $p$ and $q$). To evaluate the final auditing lower bound, we use the auditing function in (Steinke et al., 2023, Section D) for $\delta = 0$. For all methods, we use a separate set of output samples in evaluation, compared to the set of output samples used for searching for *one* optimal output set.

## 5.1 SHAPE OF OPTIMAL OUTPUT SET

We now investigate the shape of $\mathcal{O}_\tau$ for the shuffled noisy SGD under Laplace mechanism and Gaussian mechanism ((11), 12, 9, 10). Our observations are as follows.

**Optimal set contains more than two intervals for multimodal distributions** $p$ **and** $q$ In Figure 1, we observe that for mixture of Laplace or Gaussian distributions, the optimal output set contains more than two intervals. This means that the prior heuristic (Steinke et al., 2023) for selecting the top and bottom scores is strictly suboptimal. Specifically, the optimal output set $\mathcal{O}_\tau$ in Figure 1b contains the union of three disjoint intervals, one for the top-valued scores, one for bottom-valued scores, and the other for medium-valued scores. The deeper reason is that the likelihood ratio $p(o)/q(o)$ is **not monotonic** in score domain $o$ for distinguishing between general choices of distributions $p$ and $q$ that are multi-modal. See Appendix D for more examples of other distributions.

**Optimal set is asymmetric for general mechanisms** Motivated by Wilk's theorem (Wilks, 1938) (which proves that the log-likelihood ratio statistics converges asymptotically to chi-squared distribution under the null hypothesis in binary hypothesis testing), we additionally experiment on Chi-squared distributions $p$ and $q$ in Appendix D and observe that the optimal output set is asymmetric. This is because the likelihood ratio $p(o)/q(o)$ is **not symmetric** in score domain $o$ due to the asymmetric tails of chi-squared distributions $p$ and $q$. This confirms the reason for the effectiveness of the "abstaining" strategy in Steinke et al. (2023), and indicates the necessity of output set optimization for privacy auditing. See Appendix D for more examples of other output distributions.

---

[3]https://github.com/eth-sri/dp-sniper

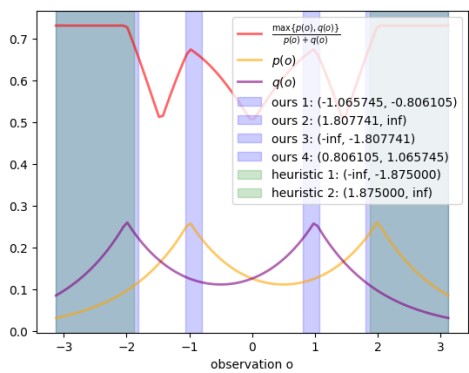

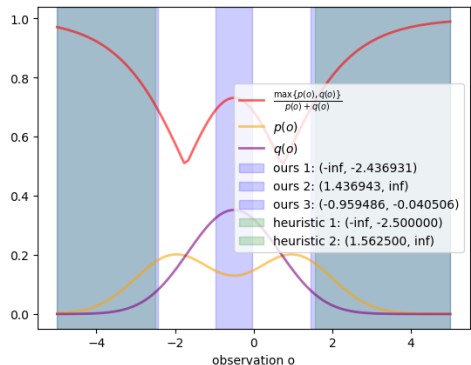

(a) Mixture of Laplace Mechanism

(b) Mixture of Gaussian Mechanism

Figure 1: Examples where the theoretical optimal output set (purple intervals) for the auditing objective (4) (given $m = 100$ output samples) contains multiple intervals. This is because the function $\frac{\max\{p(o),q(o)\}}{p(o)+q(o)}$ (red curves), which is indicative of the likelihood ratio, is non-monotonic and contains peaks in the central part of the axis. We experiment for the output distributions $p$ and $q$ in shuffled noisy SGD under *(1) Laplace mechanism* with $p = 0.5 \cdot \text{Lap}(-1, 1) + 0.5 \cdot \text{Lap}(2, 1)$ and $q = 0.5 \cdot \text{Lap}(-2, 1) + 0.5 \cdot \text{Lap}(1, 1)$, i.e., (11) and 12 with $x_1 = 4$, $x_1' = -4$, $x_2 = x_2' = 0$ and $b = 2$; *(2) Gaussian mechanism* with $p = 0.5 \cdot \mathcal{N}(-2, 1) + 0.5 \cdot \mathcal{N}(1, 1)$ and $q = 0.5 \cdot \mathcal{N}(-1, 1) + 0.5 \cdot \mathcal{N}(0, 1)$, i.e., (9) and 10 with $x_1 = -4$, $x_2 = 0$, $x_1' = -\frac{4}{3}$, $x_2' = -\frac{8}{3}$ and $\sigma^2 = \frac{16}{5}$. We also show the best-performing top and bottom output set (green intervals) that is selected according to Steinke et al. (2023), which fails to cover the central regions of the axis that have high values of $\frac{\max\{p(o),q(o)\}}{p(o)+q(o)}$.

## 5.2 Gain of Optimizing the Output Set $\mathcal{O}$

We now experimentally investigate whether choosing the optimal output set enables gain for auditing differential privacy lower bound, compared to prior (suboptimal) strategies (Bichsel et al., 2021; Lu et al., 2022; Steinke et al., 2023) for selecting the output set. Our results are summarized in Figure 2. We first observe that ***our output set optimization method consistently enables tighter privacy auditing lower bound*** than prior methods (in terms of higher mean and smaller standard deviation) across different specified guarantees $\varepsilon$, as observed in Figure 2 (left).

We also observe in Figure 2 (right) that ***our method is the most advantageous*** (compared to prior methods) ***given a small number of auditing samples*** from the output distributions $p$ and $q$. This is intuitive because when the number of samples is small, the second stochastic variance term in our optimization objective (4) is dominating. In such scenarios, it is crucial to subselect more auditing samples with the highest absolute log-likelihood ratio statistics, thus making the effect of output set optimization significant. By contrast, prior output set optimization objectives (Bichsel et al., 2021; Lu et al., 2022) do not capture such a sampling variance term, thus leading to suboptimal output sets.

Finally, we observe that the amount of ***auditing gain from the optimal output set is mechanism-dependent***. Specifically, when the number of auditing samples is large enough, all the methods lead to matching privacy lower bounds (to the upper bound $\varepsilon = 1$) in Figure 2b, thus indicating that existing auditing techniques are tight for the mixture of Laplace mechanism. However, for the mixture of Gaussian mechanism (Figure 2d), the auditing gain from our method (compared to other methods) is significant even when the number of auditing samples is large. We hypothesize that this is because mixture of Gaussian distributions incurs lighter tails than mixture of Laplace distributions, thus inducing a higher likelihood ratio in the center of the score domain and necessitating the output set selection beyond the tails. It is an interesting open problem as to what general properties of the mechanisms would make output set optimization more effective in privacy auditing.

## 6 Application: Tighter Black-box Auditing of DP-SGD

In this section, we apply our approximation method (Section 4.3) to compute the optimal output set for auditing black-box differentially private stochastic gradient descent (DP-SGD (Abadi et al.,

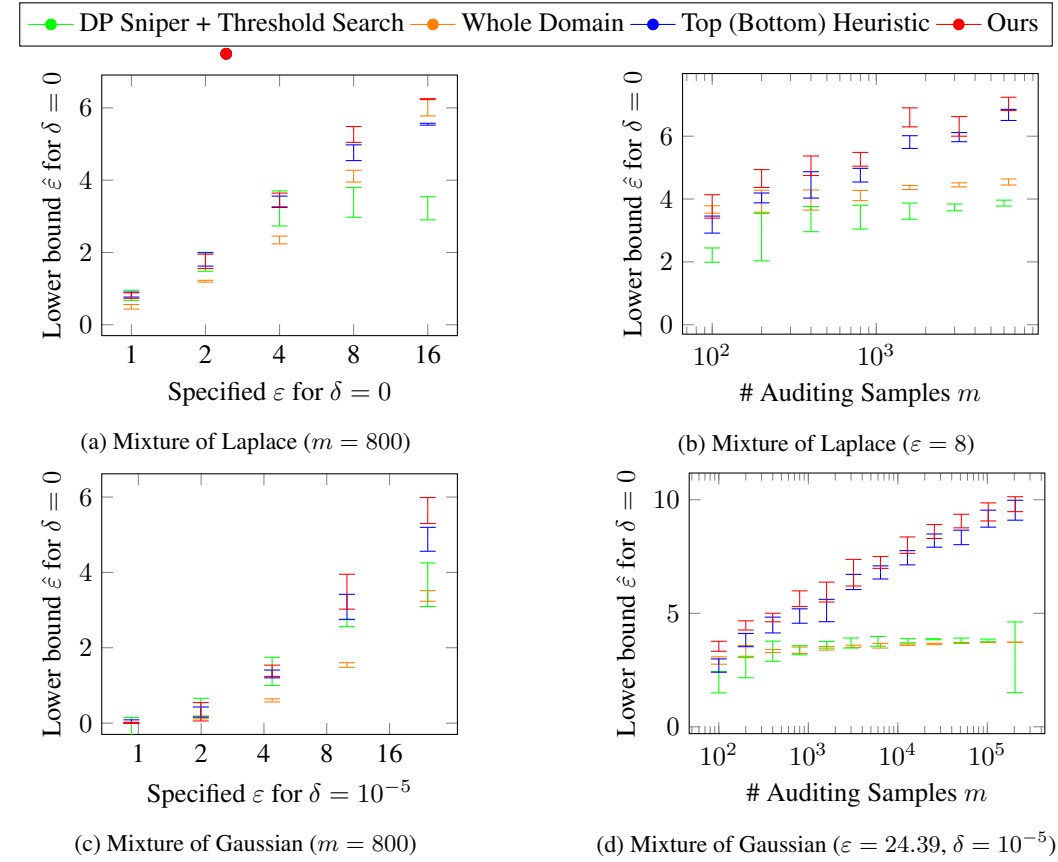

(a) Mixture of Laplace ($m = 800$)

(b) Mixture of Laplace ($\varepsilon = 8$)

(c) Mixture of Gaussian ($m = 800$)

(d) Mixture of Gaussian ($\varepsilon = 24.39, \delta = 10^{-5}$)

Figure 2: Auditing performance of our method (red) compared with prior methods (Steinke et al., 2023; Bichsel et al., 2021; Lu et al., 2022) (as discussed in Section 5). All evaluation performances are averaged across five random auditing trials. We experiment given $m \in \{100, 200, 400, 800, 1600, 3200, 6400, 12800, 51200\}$ auditing samples from the following output distributions (a) mixture of Laplace mechanism given by (11) and (12) with $x_1 = 4$, $x_1' = -4$, $x_2 = x_2' = 0$ and $b \in \{0.25, 0.5, 1, 2, 4, 8\}$ and (b) mixture of Gaussian mechanism given by (9) and (10) with $x_1 = -4$, $x_2 = 0$, $x_1' = -\frac{4}{3}$, $x_2' = -\frac{8}{3}$ and $\sigma \in \frac{4}{\sqrt{5}} \cdot \{0.25, 0.5, 1, 2, 4, 8\}$. We compute the DP guarantees $\varepsilon$ specified by different noise scale $b$ and $\sigma$ following (Dwork et al., 2006, Theorem 1) and (Balle & Wang, 2018, Theorem 8) (see Proposition C.1 for the details).

2016)). To reduce the computation cost, we focus on the one-run auditing experiment as introduced by Steinke et al. (2023). To ensure the validity of the auditing lower bound, we use the auditing function in [Steinke et al., 2023, Theorem 5.2] in our output selection Algorithm 1 (Line 6). We show that our method provides a tighter auditing lower bound than existing methods.

**Experiment Setting** We start with a state-of-the-art DP training setting from (Sander et al., 2023): running DP-SGD on the CIFAR-10 dataset with a fixed privacy budget ($\varepsilon = 8, \delta = 10^{-5}$), with 16-4-WideResNet (Zagoruyko, 2016). When given the *full* CIFAR-10 as training dataset, our experiment reaches 76% test accuracy which matches the state-of-the-art performance reported in Sander et al. (2023); De et al. (2022). Following the setting of black-box input space auditing experiment in Steinke et al. (2023, 6.2), we use loss as the MIA score, and consider the setting where the auditor only has control over the what images to use for training, cannot tweak any intermediate part (e.g., gradient) of the DP-SGD training procedure, and observes only the final trained model. Following Steinke et al. (2023), we experiment on both natural in-distribution data records (actual CIFAR-10 records) and canary data records (mislabelled CIFAR-10 records) as training datasets used for auditing. For simplicity, we focus on the setting where all records used for training are used for auditing, i.e., each record is independently included in the training dataset with half probability. This is consistent with the setting of (Steinke et al., 2023, Figure 8).

We then perform output set selection by our method and compare it with the techniques in prior works (as discussed in Section 5). To ensure that the comparison is fair, we either used the code released by the authors (for DP-sniper (Bichsel et al., 2021)) or ensured that our implementations of prior baselines enabled matching auditing performance as reported in the prior work (e.g., our observed performance for Steinke et al. (2023) in Figure 3b matches (Steinke et al., 2023, Figure 8) given $m = 10000$ auditing examples).

**Gain of choosing the optimal output set** Figure 3 summarizes our results. We show the auditing lower bounds enabled by output sets $\mathcal{O}_\tau$ constructed by our method Algorithm 1 as well as the by the heuristic intervals in Steinke et al. (2023). To rule out the effect of estimated intervals overfitting to the empirical Monte Carlo samples from output distributions, for both our method and prior methods, we used a set of fresh samples (that are disjoint from the samples used for estimating the output set) to evaluate the auditing lower bounds. We observe that our method generally provides a better (or comparable) auditing lower bound than the heuristic method in Steinke et al. (2023), in terms of higher mean or smaller variance. This shows that our method is able to approximate the optimal output set that maximizes the privacy loss lower bound, while the heuristic top (bottom) selection method in Steinke et al. (2023) may not be able to achieve this in certain settings.

**Effect of auditing data examples on the privacy lower bound** We observe that the auditing lower bound is higher for the canary dataset (Figure 3b) than the natural dataset (Figure 3a), under the same fixed specified DP guarantee. This is consistent with the intuition that the canary dataset is closer to the worst-case data record which induces high information leakage. This also suggests that the auditing lower bound is sensitive to the dataset used for auditing, and the choice of dataset can significantly affect the auditing performance. We also observe that the gain of our method is higher in the setting with 1000 records (Figure 3b) compared to the setting with 10000 records (Figure 3c). This is consistent with the intuition that the effect of selecting the optimal output set is more significant when the number of MC samples is small, as discussed in Section 5.2.

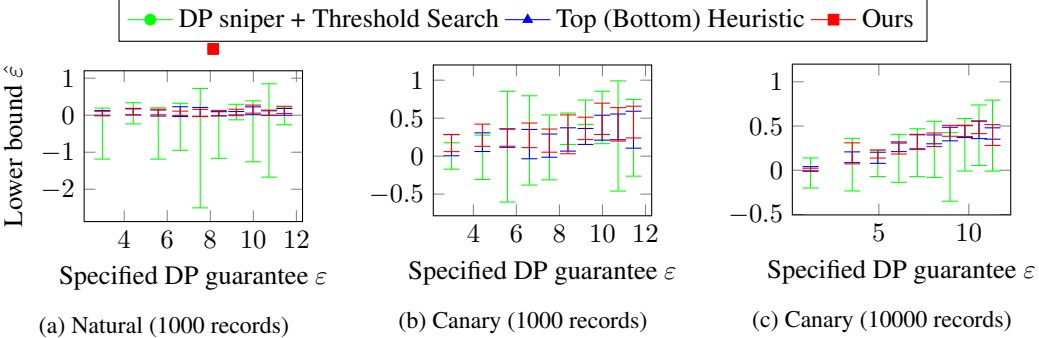

(a) Natural (1000 records)  (b) Canary (1000 records)  (c) Canary (10000 records)

Figure 3: Auditing performance versus specified DP guarantee $\varepsilon$ for our method (red) compared prior methods (Steinke et al., 2023; Bichsel et al., 2021; Lu et al., 2022) (as discussed in Section 5). We run DP-SGD on both natural in-distribution actual images and mislabelled canary images in the CIFAR-10 dataset. All performance are averaged across five random auditing trials.

# 7 CONCLUSION

In this paper, we proposed a framework to compute the optimal output event set that maximizes the privacy lower bound in privacy auditing. We derive optimality guarantee for the output set formed by thresholding likelihood ratio statistics between member and non-member score distributions in the auditing experiment. Through experiments, we show that optimizing the output set effectively tightens the privacy lower bound estimate compared to existing auditing techniques, and provides a more accurate analysis of differentially-private learning algorithms (such as subsampled Laplace and Gaussian mechanisms as well as black-box DP-SGD training). Interestingly, we find that output set optimization is the most effective when given a *restricted number of auditing examples*, or when the *likelihood ratio function is non-monotonic or assymetric* (e.g., for mechanisms with asymmetric or multi-modal output distributions). Future work includes extending our framework to other performance metrics of inference attacks, and exploring more accurate approximations of the output set when only given empirical samples from the output distributions.

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

CONTENTS

## A   DETAILS REGARDING AUDITING EXPERIMENTS IN THE LITERATURE

In this section, we explain the three components (as described in Section 3) for privacy auditing experiments in more detail.

**Computing Member and Non-member Scores** $S_+$ **and** $S_-$    Typically, the set of member scores $S_+$ and non-member scores $S_-$ consists of independent Monte Carlo samples Jagielski et al. (2020); Nasr et al. (2021; 2023); Zanella-Beguelin et al. (2023); Bichsel et al. (2021); Lu et al. (2022) from the member and non-member score distributions $p$ and $q$ specified by the auditing experiment (i.e., via computing the MIA scores on trained model in the auditing experiment and their training and test data record respectively). An exception is the recent literature of efficient privacy auditing Steinke et al. (2023); Pillutla et al. (2023), where $S_+$ and $S_-$ only consists of *weakly independent* samples due to the restricted number of trained models (one Steinke et al. (2023) or a few Pillutla et al.

| Ref | Inputs | Member scores $S_+$ | Non-member scores $S_-$ | Output set $\mathcal{O}$ | Auditing function $L$ |
|---|---|---|---|---|---|
| Jagielski et al. (2020); Nasr et al. (2021) | Datasets $D$, $D'$ with $k$ differing (poisoned) data records $D_p$. Models $\theta_1, \cdots, \theta_T \xleftarrow{i.i.d.\mathcal{T}} D$, $\theta'_1, \cdots, \theta'_T \xleftarrow{i.i.d.\mathcal{T}} D'$. | $Score(\theta_t, x)$ : $x \in D_p$ for $t = 1, \cdots, T$ | $Score(\theta'_t, x_i)$ : $x \in D_p$ for $t = 1, \cdots, T$ | $(Z, \infty)$ for threshold $Z$ | (Jagielski et al., 2020, Algorithm 2) |
| Steinke et al. (2023) | Datasets $D_{pop}$ and $D_{can}$. Model $\theta \xleftarrow{\mathcal{T}} D_{pop} \cup D$, $D \xleftarrow{\text{i.i.d. inclusion}} D_{can}$. | $Score(\theta, x_i)$ : $x_i \in D$ | $Score(\theta, x_i)$ : $x_i \in D_{can} \setminus D$ | Top $k_+$ and bottom $k_-$ values of $S_+ \cup S_-$ | (Steinke et al., 2023, Theorem 5.2), also restated in Corollary B.3 and E.2 |
| Ours | - | - | - | Algorithm 1 | - |

Table 1: Comparison between auditing experiments in the literature and our auditing experiment. Our method only modifies the construction of output set $\mathcal{O}$, and can be generally applied on top of any existing auditing experiment.

(2023)) in the auditing experiments. In this paper, we will use two standard ways to compute $S_+$ and $S_-$ in the literature Jagielski et al. (2020); Steinke et al. (2023). Our framework is general and can be applied on top of any way of computing member and non-member scores in existing auditing experiments.

**Determining the output event set $\mathcal{O}$**  To obtain higher auditing lower bound, it is crucial to choose an appropriate output set $\mathcal{O}$ for subselecting the member and non-member scores. Intuitively, this is because we would like to avoid inferring membership for outputs that are equally likely to be observed under member distribution $p$ and non-member distribution $q$. For example, when $p(o) \sim \mathcal{N}(0, 1)$ and $q(o) \sim \mathcal{N}(1, 1)$ follow Gaussian distributions, the output $o = \frac{1}{2}$ is equally likely to be observed under $p$ and $q$. By contrast, the output $o = 0$ is significantly more likely to be observed under $p$ than $q$. Thus, intuitively we select a subset $\mathcal{O}$ of member scores $S_+$ and non-member scores $S_-$ to increase their distinguishability (i.e., we would like $\mathcal{O}$ to include $o = 0$ but not $o = \frac{1}{2}$ in the Gaussian distribution example). To this end, prior works Bichsel et al. (2021); Lu et al. (2022) have proposed to train a neural network model to learn the optimal output set $\mathcal{O}$ for subselecting member and non-member scores by maximizing the likelihood difference.

However, such strategies miss an important aspect of privacy auditing experiment: $S_+$ and $S_-$ are empirical samples (rather than closed-form densities) from the output distribution. Consequently, the lower bound estimate (which is a random variable) for privacy auditing suffers from a significant sampling error when the number of selected samples from $S_+$ and $S_-$ is small. Thus, to reduce the sampling error, we would like the output set $\mathcal{O}$ to contain as many samples from $S_+$ and $S_-$ as possible.

**Designing auditing function $L$**  To design an auditing function $L$ such that Equation (1) is satisfied under $\hat{\varepsilon} := L(S_+, S_-, \mathcal{O}; \delta, \beta)$, prior works generally relies on the constraints of $(\varepsilon, \delta)$-differential privacy on the performance of a binary membership inference hypothesis test Wasserman & Zhou (2010); Kairouz et al. (2015). In this paper, we focus on an advantage-based auditing experiment, i.e., the design of the function $L$ is based on bounding the advantage of membership inference with the differential privacy guarantee $(\varepsilon, \delta)$. Extending our framework to other performance metrics of binary membership inference hypothesis tests such as true positive rate (TPR) and false positive rate (FPR) is an interesting future work.

## B  DEFERRED PROOFS FOR SECTION 4

### B.1  NOTATIONS AND RESULTS FROM PRIOR WORKS

We first define the following experiment for distinguishing between two distributions $p$ and $q$, while performing membership inference on i.i.d. samples from $p$ and $q$ that fall into a fixed output set $\mathcal{O}$.

**Definition B.1** (Experiment for distinguishing between distributions $p$ and $q$). Let $S_+$ be $m$ i.i.d. Monte Carlo samples from distribution $p$, and let $S_-$ be $m$ i.i.d. samples from distribution $q$. Let $\mathcal{O}$ be a fixed output set.

For an arbitrary inference agorithm $\mathcal{M} : \mathbb{R} \to \{-1, +1\}$ that maps an output sample $o \in \mathbb{R}$ to a binray guess in $\{-1, 1\}$ for whether the sample is drawn from $p$ (guesses 1) or $q$ (guesses $-1$), define the following experiment.

1.  For $i = 1, \cdots, m$

    (a)  Challenger samples $b_i \xleftarrow{uniform} \{1, -1\}$. If $b_i = 1$, the challenger sets $o_i$ to be the $i$-th example in $S_+$. Otherwise, the challenger sets $o_i$ to be the $i$-th example in $S_-$.

    (b)  Challenger then sends $o_i$ to the adversary.

    (c)  Adversary performs inference $\hat{b}_i = \mathcal{A}(o_i) := \begin{cases} \mathcal{M}(o_i) & \text{if } o_i \notin \mathcal{O} \\ 0 & \text{if } o_i \in \mathcal{O} \end{cases} \in \{-1, 0, 1\}$.

2.  Return $(b_i)_{i=1}^m$ and $(\hat{b}_i)_{i=1}^m$.

The above experiment is similar to the membership infererence experiment in Yeom et al. (2018, Experiment 1), except for the following differences.

1.  We used two distributions $p$ and $q$ to astract the distributions for member and non-member scores in the auditing experiment respectively. The randomness of distributions $p$ and $q$ come from many sources, such as the randomness from the dataset sampling and the output sampling of the DP mechanism. See Table 1 row one for examples of i.i.d. samples from $p$ and $q$ in prototypical auditing experiments (Jagielski et al., 2020; Nasr et al., 2021).

2.  We incorporated an output set $\mathcal{O}$ to subselect the output samples for subsequent inference. This changed the guess $\hat{b}_i$ from binary (as in Yeom et al. (2018)) to ternary, where the guess 0 indicates that the output sample $o_i$ is not in the output set $\mathcal{O}$.

To use the returned values $b_i$ and $\hat{b}_i$ from Definition B.1 to audit approximate DP, we will use the auditing bound from Steinke et al. (2023, Theorem 5.2) that generally holds for inference results in the range of $[-1, 1]$ (rather than standard auditing bounds that only holds for binary guesses, such as Yeom et al. (2018); Jagielski et al. (2020); Nasr et al. (2021)). For the simplicity of the presentation, here we restate and prove a variant of Steinke et al. (2023, Theorem 5.2) that is specialized for the output-set-dependent experiment in Definition B.1 under pure DP and i.i.d. samples.

**Lemma B.2** (Variant of Steinke et al. (2023, Proposition) under i.i.d. Monte Carlo samples). *Let $p$ and $q$ be two probability distributions over $\mathbb{R}$ that satisfies $e^{-\varepsilon} \le \frac{p(o)}{q(o)} \le e^\varepsilon$ for any $o \in \mathbb{R}$. Let $\mathcal{O} \subset \mathbb{R}$ be a fixed output set. Let $b_i, \hat{b}_i$ be defined as in experiment Definition B.1. Then conditioned on any fixed value $t \in \{-1, 0, 1\}^m$ for $(\hat{b}_i)_{i=1}^m$, it satisfies that*

$$\Pr\left[\sum_{i=1}^m \max\{b_i \cdot \hat{b}_i, 0\} \ge v \mid \hat{b} = t\right] \le \Pr_{S \leftarrow Bernoulli\left(\frac{e^\varepsilon}{e^\varepsilon + 1}\right)^m} [S_i \cdot |t_i| \ge v] \tag{13}$$

*Proof.* Observe that $b_i \cdot \hat{b}_i$ for $i = 1, \cdots, m$ are independent random variables, due to the i.i.d. sampling of $S_+, S_-$ and $b_i \xleftarrow{uniform} \{-1, 1\}$ across $i = 1, \cdots, m$ in Definition B.1. Thus to prove Equation (13), we only need to prove that

$$\frac{1}{1 + e^\varepsilon} \le \Pr[b_i = 1 \mid \hat{b}_i = t_i] \le \frac{e^\varepsilon}{1 + e^\varepsilon} \tag{14}$$

Denote $A_1 = \{o \in \mathbb{R} : \mathcal{A}(o) = 1\}$, $A_0 = \{o \in \mathbb{R} : \mathcal{A}(o) = 0\}$ and $A_{-1} = \{o \in \mathbb{R} : \mathcal{A}(o) = -1\}$ to be the preimage set of guess 1, 0 and $-1$ respectively, given membership inference strategy $\mathcal{A}$. Then by definition, we have that

$$\Pr[b_i = 1 | \hat{b}_i = t_i] = \frac{\Pr[b_i = 1, \hat{b}_i = t_i]}{\Pr[b_i = -1, \hat{b}_i = t_i] + \Pr[b_i = 1, \hat{b}_i = t_i]} \tag{15}$$

$$= \frac{\Pr[b_i = 1] \cdot \Pr[\hat{b}_i = t_i | b_i = 1]}{\Pr[b_i = -1] \cdot \Pr[\hat{b}_i = t_i | b_i = -1] + \Pr[b_i = 1] \cdot \Pr[\hat{b}_i = t_i | b_i = 1]} \tag{16}$$

$$= \frac{0.5 \cdot q(A_{t_i})}{0.5 \cdot p(A_{t_i}) + 0.5 \cdot q(A_{t_i})} \tag{17}$$

$$= \frac{1}{1 + \frac{p(A_{t_i})}{q(A_{t_i})}} \in \left[ \frac{1}{1 + e^\varepsilon}, \frac{1}{1 + e^{-\varepsilon}} \right] \tag{18}$$

where the last inequality is due to the assumed condition $e^{-\varepsilon} \le \frac{p(o)}{q(o)} \le e^\varepsilon$ for any $o \in \mathbb{R}$. $\qquad \square$

On the one hand, Lemma B.2 can be seen as a simplified variant of Steinke et al. (2023, Proposition 5.1) for auditing i.i.d. Monte Carlo samples. On the other hand, this Lemma generalizes Steinke et al. (2023, Proposition 5.1) from special designs of output set (fixed number of "abstentions") to any fixed choice of output set. This generalization then serves as the basis for proving our auditing lower bound Proposition 4.1 that is specific to the choice of output set.

As an corollary of Lemma B.2, we can obtain the following auditing function for pure DP, under subselected i.i.d. output samples that fall into output set $\mathcal{O}$.

**Corollary B.3.** *Let $\{b_i\}_{i=1}^m$ and $\{\hat{b}_i\}_{i=1}^m$ be returned by Definition B.1 under output set $\mathcal{O}$. By applying Lemma B.2 under setting $M$ to be the mechanism that maps $\{b_i\}_{i=1}^m$ to $\{\hat{b}_i\}_{i=1}^m$ in Definition B.1, we obtain following auditing function for approximate DP.*

$$L(S_+, S_-, \mathcal{O}; \delta, \beta) = \max \left\{ \varepsilon : \Pr_{S \leftarrow Bernoulli\left(\frac{e^\varepsilon}{e^\varepsilon + 1}\right)^m} \left[ S_i \cdot |\hat{b}_i| \ge v \right] \right\} \tag{19}$$

*where $v = \sum_{i=1}^m \max\{b_i \cdot \hat{b}_i, 0\}$.*

Observe that the dependence of the above auditing function (19) on the output set $\mathcal{O}$ is implicit through the dependence of $\{\hat{b}_i\}_{i=1}^m$ on $\mathcal{O}$ in Definition B.1. This makes it hard to understand the optimal choice of output set for maximizing the auditing function $L(S_+, S_-, \mathcal{O}; \delta, \beta)$. By contrast, our paper proves new auditing function in Proposition 4.1 that has explicit dependence on the output set $\mathcal{O}$, and can be used to analyze the optimal output set for auditing DP.

### B.2 PROOF FOR PROPOSITION 4.1

*Proof for Proposition 4.1.* Let $(b_i)_{i=1}^m$ and $(\hat{b}_i)_{i=1}^m$ be the outputs of experiment Definition B.1 given the $m$ i.i.d. samples $S_+$ from $p$ and $m$ i.i.d. samples $S_-$ from $q$ as inputs, and under the inference algorithm $\mathcal{M} : \mathbb{R} \to \{1, -1\}$ in Definition B.1 defined as follows.

$$\mathcal{M}(o) = \begin{cases} 1 & \text{if } o \in A \\ -1 & \text{if } o \in A^c \end{cases} \tag{20}$$

where $A = \{o \in \mathbb{R} : p(o) \ge q(o)\}$ is the acceptance region for the optimal membership inference attack that maximizes the advantage for distinguishing between $p$ and $q$ (see e.g., (Tan et al., 2022, Proposition 3.1) for the optimality guarantee).

By Lemma B.2, conditioned on any fixed $t \in \{-1, 0, 1\}^m$, the following inequality holds.

$$\Pr\left[ \sum_{i=1}^m \max\{b_i \cdot \hat{b}_i, 0\} \ge v | \hat{b} = t \right] \le \Pr_{S \leftarrow Bernoulli\left(\frac{e^\varepsilon}{e^\varepsilon + 1}\right)^m} \left[ S_i \cdot |t_i| \ge v \right] \tag{21}$$

Now by applying Hoeffding's inequality on the right-hand-side sequence of bounded random variables $S \leftarrow \text{Bernoulli}\left(\frac{e^{\varepsilon}}{e^{\varepsilon}+1}\right)^m$, we further prove that for any fixed $t \in \{-1,0,1\}^m$, it satisfies that

$$\Pr_{S \leftarrow \text{Bernoulli}\left(\frac{e^{\varepsilon}}{e^{\varepsilon}+1}\right)^m}[S_i \cdot |t_i| \geq v] \leq \exp\left(-\frac{2\left(v - \frac{e^{\varepsilon}}{e^{\varepsilon}+1} \cdot \sum_{i=1}^m |t_i|\right)^2}{\sum_{i=1}^m |t_i|}\right) \tag{22}$$

By setting $v = \frac{e^{\varepsilon}}{e^{\varepsilon}+1}\sum_{i=1}^m |t_i| + \ln\left(\frac{1}{\beta}\right)$ in (22), and by plugging the result into (13), we prove that the following inequality holds for any $t \in \{-1,0,1\}^m$.

$$\Pr\left[\sum_{i=1}^m \max\{b_i \cdot \hat{b}_i, 0\} \geq \frac{e^{\varepsilon}}{e^{\varepsilon}+1}\sum_{i=1}^m |t_i| + \sqrt{\sum_{i=1}^m |t_i| \cdot \frac{\ln(2/\beta)}{2}} \,\Big|\, \hat{b} = t\right] \leq \frac{\beta}{2} \tag{23}$$

We now prove another high probability upper bound for $\sum_{i=1}^m \max\{b_i \cdot \hat{b}_i, 0\}$ as follows. By the definition of $\hat{b}_i = \mathcal{A}(o_i)$ for $\mathcal{A}$ as defined in (20), and by Bayes rule, we have that for any $i = 1, \cdots, m$, the following equality holds.

$$\Pr[b_i \cdot \hat{b}_i = 1 | \hat{b}_i \neq 0] = \frac{\Pr[b_i = 1, S_+(i) \in \mathcal{O} \cap A] + \Pr[b_i = -1, S_-(i) \in \mathcal{O} \cap A^c]}{\Pr[b_i = 1, S_+(i) \in \mathcal{O}] + \Pr[b_i = -1, S_-(i) \in \mathcal{O}]} \tag{24}$$

$$= \frac{p(\mathcal{O} \cap A) + q(\mathcal{O} \cap A^c)}{p(\mathcal{O}) + q(\mathcal{O})} = \frac{\int_{o \in \mathcal{O}} \max\{p(o), q(o)\} do}{p(\mathcal{O}) + q(\mathcal{O})} \tag{25}$$

Thus by again using Hoeffding's inequality on the sequence of bounded random variable $b_i \cdot \hat{b}_i$ for $i = 1, \cdots, m$, we have that

$$\Pr\left[\sum_{i=1}^m \max\{b_i \cdot \hat{b}_i, 0\} \leq \frac{\int_{o \in \mathcal{O}} \max\{p(o), q(o)\} do}{p(\mathcal{O}) + q(\mathcal{O})} \cdot \sum_{i=1}^m |t_i| - \sqrt{\sum_{i=1}^m |t_i| \cdot \frac{\ln(2/\beta)}{2}} \,\Big|\, \hat{b} = t\right] \leq \frac{\beta}{2} \tag{26}$$

By combining (23) and (26) using union bound, we have that

$$\Pr\left[\frac{\int_{o \in \mathcal{O}} \max\{p(o), q(o)\} do}{p(\mathcal{O}) + q(\mathcal{O})} \cdot \sum_{i=1}^m |t_i| - \sqrt{\sum_{i=1}^m |t_i| \cdot \frac{\ln(2/\beta)}{2}} \geq \frac{e^{\varepsilon}}{e^{\varepsilon}+1}\sum_{i=1}^m |t_i| + \sqrt{\sum_{i=1}^m |t_i| \cdot \frac{\ln(2/\beta)}{2}} \,\Big|\, \hat{b} = t\right] \leq \beta$$

By rearranging the terms in the above inequality, we have that

$$\Pr\left[\varepsilon \leq \phi\left(\frac{\int_{o \in \mathcal{O}} \max\{p(o), q(o)\} do}{p(\mathcal{O}) + q(\mathcal{O})} - \sqrt{\frac{2\ln(2/\beta)}{\sum_{i=1}^m |t_i|}}\right) \,\Big|\, \hat{b} = t\right] \leq \beta \tag{27}$$

where $\phi(y) = \log\left(\frac{y}{1-y}\right)$ for $y \in (0,1)$ is the logit function. By observing that $\hat{b}_i = \mathcal{A}(o_i) \neq 0$ if any only if $o_i \in \mathcal{O}$, we have that $\sum_{i=1}^m |t_i| \leq |S_+ \cap \mathcal{O}| + |S_- \cap \mathcal{O}|$. By plugging this into (27) and taking maximum over all possible $t \in \{-1,0,1\}^m$, we obtain the bound in the statement (3). $\square$

*Remark B.4.* Proposition 4.1 generalizes prior advantage-based auditing functions Yeom et al. (2018); Steinke et al. (2023) from specific designs of output set to any fixed choice of output set. For example, the advantage-based auditing function (Yeom et al., 2018, Theorem 1) is equivalent (up to monotonic scaling) to Equation (3) under setting the whole output domain as output set, i.e., $\mathcal{O} = \mathbb{R}$. Similarly, by setting the output set $\mathcal{O} = (-\infty, o_{k_-}) \cup (o_{k_+}, +\infty)$ where $o_{k_-}$ and $o_{k_+}$ are the bottom-$k_-$ score and top-$k_+$ score in $S_- \cup S_+$, we recover the auditing function used under the abstention strategy in Steinke et al. (2023, Algorithm 1, Proposition 5.1).

*Remark B.5.* Proposition 4.1 proves an auditing function that explicitly depends on the output set $\mathcal{O}$, rather than implicitly depending on the output set as in prior works Steinke et al. (2023) (as discussed after Lemma B.2). This then allows us to analyze the optimal output set for auditing DP.

## B.3   PROOF FOR THEOREM 4.3

*Proof for Theorem 4.3.* We first prove the existence of $\mathcal{O}_\tau$ such that $p(\mathcal{O}_\tau) + q(\mathcal{O}_\tau) = p(\mathcal{O}) + q(\mathcal{O})$, for any feasible output set $\mathcal{O}$.

By observing that $p$ and $q$ are continuous output distributions on $\mathbb{R}$, we prove that the following function is continuous with respect to $\tau \geq 0$.

$$E(\tau) = p(\mathcal{O}_\tau) + q(\mathcal{O}_\tau) \text{ where } \mathcal{O}_\tau = \{x \in \mathbb{R} : \big| \log \frac{p(x)}{q(x)} \big| \geq \tau\}$$

By definition, we further have that $E(0) = 2$, $\lim_{\tau \to +\infty} E(\tau) = 0$, and $p(\mathcal{O}) + q(\mathcal{O}) \in [0, 2]$. Thus by using intermediate value theorem for continuous function, we prove that for any feasible output set $\mathcal{O}$, there exists $\tau \in \mathbb{R}$, such that $p(\mathcal{O}) + q(\mathcal{O}) = p(\mathcal{O}_\tau) + q(\mathcal{O}_\tau)$.

We now prove the optimality guarantee. Let $\mathcal{O}$ be any output set that satisfies $p(\mathcal{O}) + q(\mathcal{O}) = p(\mathcal{O}_\tau) + q(\mathcal{O}_\tau)$. By definition of $\mathcal{O}_\tau$ in Definition 4.2, we have that

$$(\mathbf{1}_{x \in \mathcal{O}_\tau} - \mathbf{1}_{x \in \mathcal{O}}) \cdot \left( \max\{p(x), q(x)\} - \frac{e^\tau}{1 + e^\tau} \cdot (p(x) + q(x)) \right) \geq 0 \tag{28}$$

holds for any $x \in \mathbb{R} \subset A$ where $A$ is a ignorable set.

Doing the integration over $x \in \mathbb{R}$, we immediately have that

$$\int_{x \in \mathcal{O}_\tau} \max\{p(x), q(x)\} dx - \frac{e^\tau}{1 + e^\tau} \cdot (p(\mathcal{O}_\tau) + q(\mathcal{O}_\tau)) \tag{29}$$

$$\geq \int_{x \in \mathcal{O}} \max\{p(x), q(x)\} dx - \frac{e^\tau}{1 + e^\tau} \cdot (p(\mathcal{O}) + q(\mathcal{O})) \tag{30}$$

By plugging the condition that $p(\mathcal{O}) + q(\mathcal{O}) = p(\mathcal{O}_\tau) + q(\mathcal{O}_\tau)$ into the constraints, we have that

$$\int_{x \in \mathcal{O}_\tau} \max\{p(x), q(x)\} dx \geq \int_{x \in \mathcal{O}} \max\{p(x), q(x)\} dx \tag{31}$$

Thus by definition of auditing bound for output set in Equation (4), we have that $\hat{\varepsilon}(\mathcal{O}_\tau; p, q) \geq \hat{\varepsilon}(\mathcal{O}; p, q)$. □

*Remark* B.6 (Similarity in proof technique to Neyman-Pearson Lemma). The proof technique, which constructs an indicator function that is always non-negative (eq 21), and then performs integration (eq 22 and 23), is the standard technique used for poving Neyman-Pearson Lemma.

## C   DEFERRED PROOFS FOR SECTION 5

**Proposition C.1.** *Let $b_1, b_2, \sigma_1, \sigma_2 \geq 0$, $\theta_0 \in \mathbb{R}$ and $\eta \leq 1$ be fixed. Assume that the input dataset $D$ has bounded domain, in that there exists $r \geq 0$ such that $|x| \leq r$ for any $x \in D$. Define division over zero as infinity, i.e., $\frac{1}{0} = \infty$. Then*

- *If $Z_1 \sim Lap(0, b_1)$ and $Z_2 \sim Lap(0, b_2)$, then the mechanism with output distribution (8) satisfies $\varepsilon$-DP with $\varepsilon = \frac{r\eta(2-\eta)}{\max\{\eta(1-\eta)b_1, \eta b_2\}}$.*

- *If $Z_1 \sim \mathcal{N}(0, \sigma_1^2)$ and $Z_2 \sim \mathcal{N}(0, \sigma_2^2)$, then the mechanism with output distribution (8) satisfies $(\varepsilon, \delta)$-DP with $\delta = \bar{\Phi}\left(\frac{r(2-\eta)}{2\sigma} - \frac{\varepsilon\sigma}{r(2-\eta)}\right) - e^\varepsilon \cdot \bar{\Phi}\left(-\frac{r(2-\eta)}{2\sigma} - \frac{\varepsilon\sigma}{r(2-\eta)}\right)$ for any $\varepsilon \geq 0$, where $\sigma = \sqrt{(1-\eta)^2\sigma_1^2 + \sigma_2^2}$ and $\Phi$ denotes the cumulative distribution function of the standard normal distribution.*

*Proof.* It suffices to apply (Dwork et al., 2006, Theorem 1) and (Balle & Wang, 2018, Theorem 8) after observing that the $\ell_1$ and $\ell_2$ sensitivity of the update in (8) is $\max_{x_{s_1}, x_{s_2}} |-\eta \cdot (1 - \eta)x_{s_1} + \eta x_{s_2}| \leq \eta(2 - \eta) \cdot r$ for $\eta \leq 1$. □

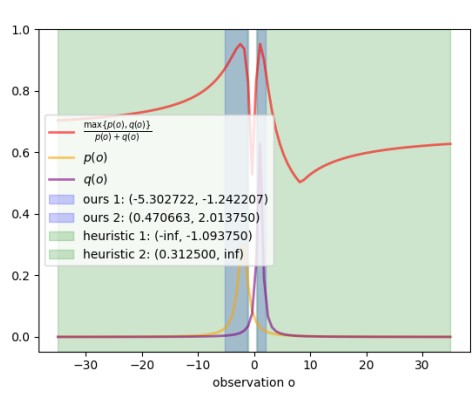
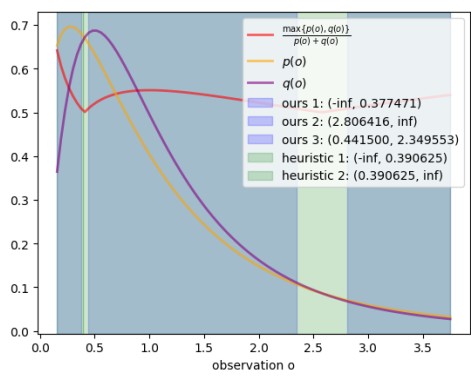



(a) Two Cauchy distributions          (b) Two F distributions



Figure 4: Examples where our selected optimal output set (purple intervals) is asymmetric and consists of multiple intervals, given a small number of $m = 100$ auditing MCMC samples. We show the function $\frac{\max\{p(o),q(o)\}}{p(o)+q(o)}$ (that is indicative of the absolute value of log-likelihood ratio) for the output distributions $p$ and $q$ in shuffled noisy SGD under **(1) Cauchy distributions** with $p = \text{Cauchy}(-1, 0.1)$ and $q = \text{Cauchy}(1, 3)$; **(2) F distributions** with $p = \text{F}(3, 10)$ and $q = \text{F}(5, 10)$. We also show the best-performing top and bottom output set (green intervals) that is selected according to Steinke et al. (2023), which fails to cover the central regions of the axis that have high $\frac{\max\{p(x),q(x)\}}{p(x)+q(x)}$.

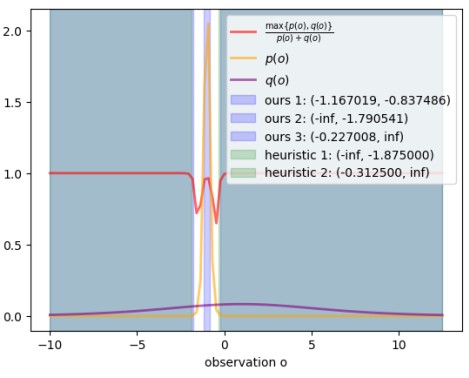
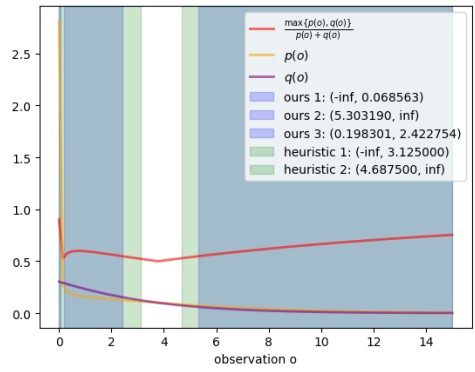



(a) Two Logistic distributions          (b) Two Chi-squared distributions



Figure 5: Examples where our selected optimal output set (purple intervals) is asymmetric and consists of multiple intervals, given a small number of $m = 100$ auditing MCMC samples. We show the function $\frac{\max\{p(o),q(o)\}}{p(o)+q(o)}$ (that is indicative of the absolute value of log-likelihood ratio) for the output distributions $p$ and $q$ in shuffled noisy SGD under **(1) Logistic distributions** with $p = \text{Logistic}(-1, 0.1)$ and $q = \text{Logistic}(1, 3)$; **(2) Chisquared distributions** with $p = \chi_1^2(3)$ and $q = \chi_2^2(1)$, i.e., $p$ follows a non-central Chi-squared distribution with 1 degree of freedom and non-centrality parameter 3, while $q$ has 2 degrees of freedom and non-centrality parameter 1. We also show the best-performing top and bottom output set (green intervals) that is selected according to Steinke et al. (2023), which fails to cover the central regions of the axis that have high $\frac{\max\{p(x),q(x)\}}{p(x)+q(x)}$.

# D    MORE EXAMPLES FOR THE SHAPE OF OUTPUT SET IN OTHER MECHANISMS

# E    ADDITIONAL RESULTS

## E.1    APPLYING ALGORITHM 1 TO AUDIT APPROXIMATE DIFFERENTIAL PRIVACY

In this section, we give examples of applying our output set selection Algorithm 1 to audit approximate $(\varepsilon, \delta)$-differential privacy. To ensure that the auditing lower bound is valid for approximate DP, in Line 6 of Algorithm 1, we need to use the following bound from Steinke et al. (2023, Theorem 5.2) that is valid for auditing approximate DP using subselected output scores.

**Theorem E.1.** *Steinke et al. (2023, Theorem 5.2) Let $M : \{-1, +1\}^m \to [-1, +1]^m$ satisfy $(\varepsilon, \delta)$-DP. Let $T = M(S) \in [-1, +1]^m$. Then, for any $v \in \mathbb{R}$,*

$$\Pr_{S \leftarrow \{-1, +1\}^m, T \leftarrow M(S)} \left[ \sum_{i=1}^m \max\{0, T_i \cdot S_i\} \geq v \right] \leq \beta(\varepsilon) + \alpha(\varepsilon) \cdot 2m \cdot \delta \qquad (32)$$

*where*

$$\beta(\varepsilon) = \Pr_{\check{W}^*} \left[ \check{W}^* \geq v \right], \qquad (33)$$

$$\alpha(\varepsilon) = \max \left\{ \frac{1}{i} \left( \Pr_{\check{W}^*} [\check{W}^* \geq v - i] - \beta(\varepsilon) : i \in \{1, 2, \cdots, m\} \right) \right\}. \qquad (34)$$

*Here $\check{W}^*$ is any distribution on $\mathbb{R}$ that stochastically dominates $\check{W}(t) := \sum_{i=1}^m \check{S}_i |t_i|$ for $\check{S} \leftarrow$ Bernoulli $\left( \frac{e^\varepsilon}{e^\varepsilon + 1} \right)^m$ for all $t$ in the support of $T$.*

By applying Theorem E.1 to Definition B.1, we can obtain the following auditing function for approximate DP under subselected scores from output distributions by output set $\mathcal{O}$.

**Corollary E.2** (Auditing function for approximate DP)**.** *Let $\{b_i\}_{i=1}^m$ and $\{\hat{b}_i\}_{i=1}^m$ be as defined in Definition B.1 under output set $\mathcal{O}_\tau$. By applying Theorem 4.3 under setting $M$ to be the mechanism that maps $\{b_i\}_{i=1}^m$ to $\{\hat{b}_i\}_{i=1}^m$, we obtain following auditing function for approximate DP.*

$$L(S_+, S_-, \mathcal{O}_\tau; \delta, \beta) = \max \{\varepsilon : \beta(\varepsilon) + \alpha(\varepsilon) \cdot 2m \cdot \delta \leq \beta\} \qquad (35)$$

*where $\beta(\varepsilon)$ and $\alpha(\varepsilon)$ are defined in Theorem E.1 under setting $v = \sum_{i=1}^m \max\{b_i \cdot \hat{b}_i, 0\}$.*

**Auditing Performance for Approximate DP**    Our results are summarized in Figure 6. We observe that our method outperforms the prior methods (Steinke et al., 2023) and the whole domain baseline in terms of auditing performance. We also observe that the audited lower bound $\hat{\varepsilon}$ monotonically decreases as $\delta$ increases, which is consistent with the theoretical guarantee (dashed line in Figure 6).

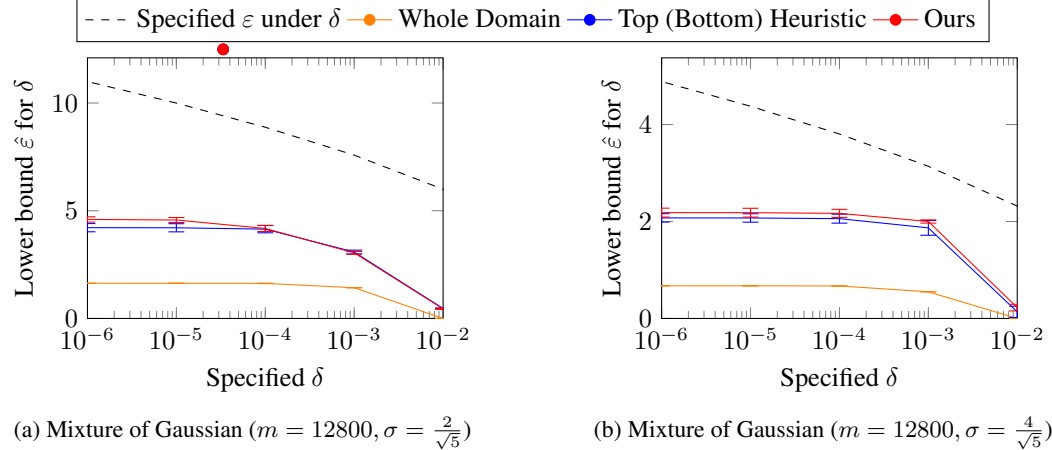

(a) Mixture of Gaussian ($m = 12800, \sigma = \frac{2}{\sqrt{5}}$)

(b) Mixture of Gaussian ($m = 12800, \sigma = \frac{4}{\sqrt{5}}$)

Figure 6: Auditing performance (for approximate DP) of our method (red) compared with prior method (Steinke et al., 2023) and whole domain baseline (as discussed in Section 5). All evaluation performances are averaged across five random auditing trials. We experiment given $m = 12800$ auditing samples from the output distribution of mixture of Gaussian mechanism given by (9) and (10) with $x_1 = -4$, $x_2 = 0$, $x'_1 = -\frac{4}{3}$, $x'_2 = -\frac{8}{3}$ and $\sigma \in \frac{4}{\sqrt{5}} \cdot \{0.5, 1\}$. We compute the DP guarantees $\varepsilon$ for $\delta \in \{10^{-6}, \cdots, 10^{-2}\}$ specified by different noise scales $\sigma$ following (Dwork et al., 2006, Theorem 1) and (Balle & Wang, 2018, Theorem 8) (see Proposition C.1 for the details).

