# OpenReview forum: "Identifying Optimal Output Sets for Differential Privacy Auditing"
_ICLR.cc/2025/Conference — Submitted to ICLR 2025_

### Official Review · Reviewer_DqzP · 2024-11-01

**Soundness:** 3
**Presentation:** 3
**Contribution:** 3
**Rating:** 6
**Confidence:** 3

**Summary:**

This paper introduces a novel framework for privacy auditing. The framework leverages the likelihood ratios function thresholding to select the optimal output set, which maximizes the privacy loss lower bound. The optimality of the proposed framework is proved and the advantage over existing approaches is validated by empirical results.

**Strengths:**

The result is proved to be optimal as stated in Theorem 4.3. This theoretical finding is supported by empirical evidence presented in Sections 5 and 6.

**Weaknesses:**

(1) The proposed framework builds upon existing techniques such as likelihood ratio thresholding.

(2) Some implementation details are unclear to me. I include them in the questions (1) and (2).

**Questions:**

(1) Can the author provide more details on how to sample from p and q for the experiments presented in section 6?

(2) What's the size of levels $\tau$ ?

(3) In proposition 4.1, what's the purpose of setting $\delta=0$? Does it only apply to auditing pure-DP?

(4) In algorithm 1, samples drawing from p and q are assumed to be equal. Is this assumption needed?

---

> ### Author Response · Authors · 2024-11-22
>
> Thanks for the valuable feedbacks. Below we answer the questions.
> > Can the author provide more details on how to sample from p and q for the experiments presented in section 6?
>
> We follow the one-run auditing experiment in [Algorithm 1] to generate the member scores and non-member scores. Below we restate the sampling process from [Steinke et al., 2023, Algorithm 1] in our notations, and our simplified experiment setting of $m=n$.
>
> 1. **Data:** $x\in \mathcal{X}^m$ consisting of $m$ auditing samples. Training algorithm $\mathcal{T}$
> 2. For $i\in [m]$ samples $b_i\xleftarrow{uniform}\{-1, 1\}$ independently.
> 3. Partition $x$ into $x_{IN}\in\mathcal{X}^{n_{IN}}$ and $x_{OUT}\in\mathcal{X}^{n_{OUT}}$ according to $b$, where $n_{IN} + n_{OUT} = n$. Namely, if $b_i=1$, then $x_i$ is in $x_{IN}$; and, if $b_i=-1$, then $x_i$ is in $x_{OUT}$.
> 4. Run $\mathcal{T}$ on input $x_{IN}$ with appropriate parameters, output model $\theta$.
> 5. Compute the vector of member scores $S_+=(SCORE(x_i, \theta): x_i\in x_{IN})$, and the vector of non-member scores $S_-=(SCORE(x_i, \theta): x_i\in x_{OUT})$
> 6. Return: $S_+=(SCORE(x_i, \theta): x_i\in x_{IN})$ and $S_-=(SCORE(x_i, \theta): x_i\in x_{OUT})$
>
>
> The returned member and non-member score samples $S_+$ and $S_-$ can then be used as inputs for our output set selection Algorithm 1.
>
> > What's the size of levels $\tau$?
>
> We have updated Algorithm 1 (Line 3) to reflect the set of levels $\tau$ that we search over, which contains $2m+1$ values with each reflecting the log-likelihood-ratio on the interval between the $i$-th largest output sample and the $i + 1$-th largest output sample in $S_+\cup S_-$.
>
> In experiments, when the number of output samples is large, we would only evaluate a subset of the $2m$-log likelihood-ratio levels (e.g., only evaluating $\tau_k, \tau_{2k}, \cdots$ for $k>1$) to improve the efficiency of running Algorithm 1.
>
>
> > In proposition 4.1, what's the purpose of setting $\delta=0$? Does it only apply to auditing pure-DP?
>
> Proposition 4.1 indeed only establishes auditing bound for pure DP. This is for simplicity of presentation, as the auditing lower bound for pure DP (Corollary B.3) takes a significantly simpler form than that auditing lower bound for approximate DP (Corollary E.2).
>
> However, our framework readily adapts to approximate DP auditing,  as long as the auditing function used in the score set selection step (Line 6 in Algorithm 1) is valid for $\delta>0$. **As an example, we have added the results for auditing approximate DP for the mixture of Gaussian mechanisms in Appendix E.1 of the revised paper.**
>
> > In algorithm 1, samples drawing from p and q are assumed to be equal. Is this assumption needed?
>
> This is also for simplicity of presentation. Algorithm 1 is also applicable to the setting where the number of samples from $p$ and $q$ are not equal. We have updated Algorithm 1 to reflect this applicability.

---

### Official Review · Reviewer_4UzV · 2024-11-03

**Soundness:** 2
**Presentation:** 2
**Contribution:** 2
**Rating:** 6
**Confidence:** 4

**Summary:**

This paper studies privacy auditing for differential privacy. The paper proposes an approach that claims to identify or approximate the optimal output event sets that can achieve maximal privacy loss lower bound in auditing.

**Strengths:**

The paper studies the problem of identifying optimal output event sets for differential privacy auditing, which is an important problem that can lead to better and more efficient privacy auditing.

**Weaknesses:**

However, the paper has significant flaws in both the theoretical claims and the methodological support for its core assertions. Please refer to the Questions below for more details.

I am open to increasing my score if the authors can provide convincing clarifications or solutions to my concerns in the rebuttal.

**Questions:**

Comment 1:

Theorem 4.3 cannot prove that the $\tau$-log-likelihood-ratio-set enables maximum auditing lower bound objective (4) among all possible choices of output set $S$.  I will explain this as follows.

First, by fixing p and q, there always exists a $\tau'$ for any feasible output set $\mathcal{O}$ such that this $\mathcal{O}$ is $\tau'$ -log-likelihood-ratio-set; i.e.,  $\mathcal{O}=\mathcal{O}$\_$\tau'$.

Now, in Section B.2 PROOF FOR THEOREM 4.3, let's replace $\mathcal{O}$_$\tau$ by $\mathcal{O}$\_$\tau'$, and replace $\tau$ by $\tau'$. In addition, let's replace $\mathcal{O}$ by any $\mathcal{O}'$ that satisfies $p(\mathcal{O}')$ + $q(\mathcal{O}')$ = p($\mathcal{O}$\_$\tau'$) + q($\mathcal{O}$\_$\tau'$) .

By following the same steps, we can obtain the similar inequality of Eq (24), where the left-hand side integration is over the set $\mathcal{O}$\_$\tau'$ and the right-hand side integration is over the set $\mathcal{O}'$ for all $\mathcal{O}'$ satisfying $p(\mathcal{O}')$ + $q(\mathcal{O}')$ = p($\mathcal{O}$\_$\tau'$) + q($\mathcal{O}$\_$\tau'$). Let's call this inequality as Virtual-Eq (24).

Since the original setting in the paper is p($\mathcal{O}$\_$\tau'$) + q($\mathcal{O}$\_$\tau'$) ) = p($\mathcal{O}$\_$\tau$) + q($\mathcal{O}$\_$\tau$) (recall that  $\mathcal{O}=\mathcal{O}$\_$\tau'$), it is obvious that $\mathcal{O}$\_$\tau$ is one of $\mathcal{O}'$ that satisfies $p(\mathcal{O}')$ + $q(\mathcal{O}')$ = p($\mathcal{O}$\_$\tau'$) + q($\mathcal{O}$\_$\tau'$).

Therefore, from the original Eq (24) and the Virtual-Eq (24), we obtain the following:

Integral-over-$\mathcal{Q}$\_$\tau$ max\{p(x), q(x)\} dx = Integral-over-$\mathcal{Q}$ max\{p(x), q(x)\} dx, for all $\mathcal{Q}$ satisfying  $p(\mathcal{O})$ + $q(\mathcal{O})$ = p($\mathcal{O}$\_$\tau$) + q($\mathcal{O}$\_$\tau$).

That is, Eq (24) holds only at equality, and it cannot imply $\hat{\epsilon}$ ($\mathcal{O}$\_$\tau$; p, q) $\geq$ $\hat{\epsilon}$ ($S$ p, q) for all possible choices of output set $S$.

Hence, the conclusion given by Section 4.2 IDENTIFYING OPTIMAL OUTPUT SET FOR AUDITING is incorrect. The paper does not provide the theoretical claims or methodological support for the assertion that the proposed approach can identify or approximate the optimal output set.

Comment 2:

Even if Theorem 4.3 shows some reasonable inequality-based conclusion, the conclusion only applies for the output set satisfying $p(\mathcal{O})$ + $q(\mathcal{O})$ = p($\mathcal{O}$\_$\tau$) + q($\mathcal{O}$\_$\tau$) for a given tau, and cannot be directly generalized to all possible output set $S$.


Comment 3:

In addition, the choice $\tau$ of the proposed approach seems to be heuristic or arbitrary. That is, the paper does not show how to choose $\tau$. Since the $\tau$-log-likelihood-ratio-set output set $\mathcal{O}$\_$\tau$ depends on the choice of the threshold $\tau$, the optimality of $\mathcal{O}$\_$\tau$ in general depends on $\tau$. Any related threshold-based results, claiming to be optimal without characterizing the optimality of the $\tau$, is problematic and not rigorous.

Other Comments:

Is $p(\mathcal{O})$ + $q(\mathcal{O})$ = $\tau$ above Theorem 4.3. a typo?

It is unclear how the proof of Theorem 4.3 is related to the Neyman-Pearson lemma.

If the mechanism is the training process of a machine-learning model, then does each empirical sample used in Algorithm 1 require a run of the training process? The authors should discuss the related computational costs and complexity to approximate the densities.

---

> ### Author Response · Authors · 2024-11-22
> **Response to questions [comment 1-comment 3] and other comments [O1-O2]**
>
> > [Comment 1] First, by fixing p and q, there always exists a $\tau'$ for any feasible output set $\mathcal{O}$ such that this $\mathcal{O}$ is  $\tau'$-log-likelihood-ratio-set; i.e., $\mathcal{O} = \mathcal{O}_{\tau'}$. ...
>
> To our understanding, the reviewer is trying to prove a reverse direction inequality of eq (24) and use it to prove that the inequality in Theorem 4.3 only holds at equality, thus contradicting the optimality guarantee.
>
> However, we believe there is confusion regarding the definition of $\tau'$-log-likelihood-ratio-set -- most feasible output set $\mathcal{O}$ cannot be represented by a $\tau'$-log-likelihood-ratio-set. That is, there does not exist $\tau'\in\mathbb{R}$ such that $\mathcal{O}=\mathcal{O}\_{\tau'}$.
>
> To see this, we use Guassian density $p\sim\mathcal{N}(0, 1)$ and $q\in\mathcal{N}(1,1)$ as an example. The log-likelihood ratio is
> $$\log\frac{p(x)}{q(x)} = - \frac{x^2}{2} + \frac{(x-1)^2}{2} = \frac{-2x+1}{2}$$
>
> Thus, by Definition 4.2, for any $\tau\geq 0$ the set $\mathcal{O}\_\tau$ is as follows.
> $$\mathcal{O}\_\tau = \\{x\in\mathbb{R}: \Big|\frac{-2x+1}{2}\Big| \geq \tau \\} = \Big(-\infty, \frac{1}{2} - \tau\Big]\cup\Big[\frac{1}{2} + \tau, +\infty\Big)$$
> That is, a $\tau$-log-likelihood-ratio set is always a combination of two intervals that are symmetric across the vertical $x = \frac{1}{2}$. Consequently, for most output sets $\mathcal{O}$, such as $\mathcal{O}=(-\frac{1}{3}, \frac{1}{3})$, we have that $\mathcal{O}\neq \mathcal{O}\_{\tau'}$ for any $\tau'$. That is, there does not exist $\tau'$ such that $\mathcal{O}\_{\tau'} = \mathcal{O}$.
>
>
> We are happy to address any follow-up questions, or clarifications if we misunderstood the reviewer's comment.
>
> > [Comment 2] Even if Theorem 4.3 shows some reasonable inequality-based conclusion, the conclusion only applies for the output set satisfying $p(\mathcal{O}) + q(\mathcal{O}) = p(\mathcal{O}\_\tau) + q(\mathcal{O}\_\tau)$ for a given tau, and cannot be directly generalized to all possible output set $S$.
>
> **Optimality guarantee established by Theorem 4.3** Theorem 4.3 essentially proves that for any output set $\mathcal{O}$, there exists a $\tau$-log-likelihood-ratio-set $\mathcal{O}\_\tau$ that satisfies $p(\mathcal{O}\_\tau) + q(\mathcal{O}\_\tau) = p(\mathcal{O}) + q(\mathcal{O})$ such that  $\mathcal{O}\_{\tau}$ enables higher auditing lower bound objective [Eq 4] than $\mathcal{O}$. Consequently, the family of $\\{\mathcal{O}\_\tau\\}_{\tau>0}$ are the optimal output sets for privacy auditing.
>
> We have updated the text below Theorem 4.3 in the paper to clarify more about this optimality guarantee.
>
> > [Comment 3] In addition, the choice $\tau$ of the proposed approach seems to be heuristic or arbitrary. That is, the paper does not show how to choose $\tau$. Since the $\tau$-log-likelihood-ratio-set output set $\tau$ depends on the choice of the threshold $\tau$, the optimality of $\mathcal{O}_\tau$ in general depends on $\tau$. Any related threshold-based results, claiming to be optimal without characterizing the optimality of the $\tau$, is problematic and not rigorous.
>
> The reviewer is correct that our optimality guarantee holds for the family of $\tau$-log-likelihood-ratio-set for $\tau\geq 0$, rather than for a specific choice of $\tau$. Therefore, to choose one single output set over the family of $\mathcal{O}_\tau$, we need to additionally search for the optimal threshold $\hat{\tau}$. This is a one-dimensional optimization problem over $\tau\in\mathbb{R}$, which is significantly easier and incurs significantly less computation cost than the original output set optimization problem over all possible output sets $\mathcal{O}\subseteq \mathbb{R}$.
> - When distributions $p$ and $q$ are known densities, the optimal $\tau$ can be analytically solved via computing the $\tau$-log-likelihood-ratio-set analytically and plugging it into our optimization objective [Eq 4].
> - When distributions $p$ and $q$ are unknown, we use their KDE approximations to optimize the threshold -- we have updated Algorithm 1 Line 6 to precisely reflect how we search for the threshold $\tau$.
>
>
> > [Other comments][O1] Is $p(\mathcal{O}) + q(\mathcal{O}) = \tau$ above Theorem 4.3. a typo?
>
> Yes, thanks for pointing out. We have corrected it in revised paper to be $p(\mathcal{O}) + q(\mathcal{O}) = p(\mathcal{O}\_\tau) + q(\mathcal{O}\_\tau)$.
>
> >[Other comments][O2] It is unclear how the proof of Theorem 4.3 is related to the Neyman-Pearson lemma.
>
> The proof technique, which constructs an indicator function that is always non-negative (eq 21), and then performs integration (eq 22 and 23), is the standard technique used for poving Neyman-Pearson Lemma. (E.g., see the [wikipedia of Neyman-Pearson Lemma](https://en.wikipedia.org/wiki/Neyman–Pearson_lemma) -- proof for existence.)

---

> > ### Comment · Reviewer_4UzV · 2024-11-22
> > **Comments to authors' responses**
> >
> > > **AR2: Optimalit guarantee stablished by Theorem 4.3** Theorem 4.3 essentially proves that for any output set $\mathcal{O}$, there exists a $\tau$-log-likelihood-ratio-set $\mathcal{O} _\tau$ that satisfies $p(\mathcal{O} _\tau) + q(\mathcal{O} _\tau) = p(\mathcal{O}) + q(\mathcal{O})$ such that ....
> >
> > **Comment 2:**
> >
> > This statement (and the one highlighted below Theorem 4.3 in the revised paper) is incorrect. In fact, **Theorem 4.3 does not prove the existence of $\mathcal{O} _\tau$.** Rather, Theorem 4.3 assumes the existence of $\mathcal{O} _\tau$ as a necessary condition. Specifically, the theorem states "**Given a fixed $\tau$**, if $\mathcal{O} _\tau$ is the $\tau$-log-likelihood-ratio-set for ..., then ...". Additionaly, the proof of Theorem 4.3 shows that the existence is indeed a necessary condition.
> >
> >
> > Furthermore, the conclusion of Theorem 4.3 applies **only to a fixed $\tau$.** The statement below "That is, the family of $\{\mathcal{O} _{\tau}\} _{\tau>0}$ are the optimal output sets for privacy auditing." is also incorrect.
> >
> >
> > First, the Theorem 4.3 does not establish or prove this statement. Second, this family of $\{\mathcal{O} _{\tau}\} _{\tau>0}$ excludes subsets corresponding to $\tau = 0$; if this statement were true, it would imply that most subsets are optimal, since every subset has a corresponding $\tau'$ satisfying the inequality of Equation (5).
> >
> >
> > > AR3: The reviewer is correct that our optimality guarantee holds for the family of $\tau$--log-likelihood-ratio-set for $\tau\geq 0$, rather than for a specific choice of $\tau$. Therefore, ...., we need to additionally search for **the optimal threshold** $\hat{\tau}$. ...
> >
> > Comment 3:  The authors explicitly state in this response that theoptimality guarantee requires an additional serach for the optimal threshold $\hat{\tau}$, which implies that only $\mathcal{O} _{\hat{\tau}}$ (corresponding to the optimal $\hat{\tau}$) is guaranteed to be optimal. **This contradicts the claim** "That is, the family of $\{\mathcal{O} _{\tau}\} _{\tau>0}$ are the optimal output sets for privacy auditing."
> >
> >
> >
> >
> >
> > > AR4: The proof technique, which constructs an indicator function that is always non-negative (eq 21),... the standard technique used for poving Neyman-Pearson Lemma.
> >
> > **Comment 4:** The statement "The proof (Appendix B.2) is similar to the Neyman-Pearson lemma" remains unclear and potentially misleading. It is true that the use of an indicator function and integration is a standard proof technique. However, it is a generic method that is not specific to the Neyman-Pearson lemma. Referring to this technique as a justification for similarity oversimplifies the structural and conceptual differences between the two.
> >
> > For clarity, the authors should explicitly specify whether the similarity refers to the result, methodology, or a specific aspect of the Neyman-Pearson lemma, rather than relying on a vague comparison.

---

> > > ### Author Response · Authors · 2024-11-23
> > > **Response to follow-up comment 2-4**
> > >
> > > > This statement (and the one highlighted below Theorem 4.3 in the revised paper) is incorrect. In fact, Theorem 4.3 does not prove the existence of $\mathcal{O}_\tau$...
> > >
> > > We have modified the statement of Theorem 4.3 to include the existence of $\mathcal{O}_\tau$ (as explained in the above comment), and have added its proof in Appendix B.2.
> > >
> > >
> > > > Furthermore, the conclusion of Theorem 4.3 applies only to a fixed $\tau$. The statement below "That is, the family of $\mathcal{O}_{\\{\tau>0\\}}$ are the optimal output sets for privacy auditing." is also incorrect.
> > >
> > > > The authors explicitly state in this response that theoptimality guarantee requires an additional serach for the optimal threshold ... This contradicts the claim "That is, the family of $\mathcal{O}_{\\{\tau>0\\}}$ are the optimal output sets for privacy auditing."
> > >
> > > We have corrected the typo $\\{\mathcal{O}\_\tau\\}\_{\tau>0}$ to $\\{\mathcal{O}\_\tau\\}\_{\tau>0}$.
> > > We agree that simply referring to Theorem 4.3 as ``the family of $\mathcal{O}\_{\\{\tau>0\\}}$ are the optimal output sets'' could be vague, and misleading depending on how one interpret optimal. We have updated it to the following more precise statement in the revised paper:
> > >
> > > "Theorem 4.3 proves that the family of $\tau$-log-likelihood-ratio-set **contains** the optimal output set."
> > >
> > >
> > > > The statement "The proof (Appendix B.2) is similar to the Neyman-Pearson lemma" remains unclear and potentially misleading. ... For clarity, the authors should explicitly specify whether the similarity refers to the result, methodology, or a specific aspect of the Neyman-Pearson lemma, rather than relying on a vague comparison.
> > >
> > > Thanks for the suggestion, we have added the similarity discussion in Remark B.6 in the Appendix, and have modified the statement in the main paper to be
> > >
> > > "
> > > The proof technique (Appendix B.2) is similar to the Neyman-Pearson Lemma (Neyman & Pearson, 1933) (Remark B.6).
> > > "

---

> ### Author Response · Authors · 2024-11-22
> **Response to Other comments [O3]**
>
> > [O3] If the mechanism is the training process of a machine-learning model, then does each empirical sample used in Algorithm 1 require a run of the training process? The authors should discuss the related computational costs and complexity to approximate the densities.
>
> For the auditing training process, if one were to require i.i.d. output samples, then indeed each empirical sample would require a fresh run of the training process, as done in prototypical auditing experiments (Jagielski et al., 2020; Nasr et al., 2021). Due to the huge computation cost, we do not use such an auditing experiment in Section 6.
>
>
> Instead, we use the recent auditing experiment in [steinke2023] that only requires one run of the training algorithm -- where each empirical sample would be the score of the trained model on one "canary" data record, and one can obtain many samples by evaluating the score of one trained model on the whole "canary" dataset. The "trick" is to randomize the inclusion of each "canary" data into the training dataset in the auditing experiment. By carefully taking the correlation between MIA guesses for different data records into consideration, [Theorem 5.2, steinke2023] proves auditing lower bound under such a one-run setting.
>
> We have updated Algorithm 1 to more precisely reflect the comptuation cost for KDE estimation.
>
>
> - The computation cost for KDE estimator in Algorithm 1 (Line 1) require two runs of KDE estimation on $m$ auditing samples, which takes less than a minute on a standard computater when $m$ is in the order $10^6$.
>
> - The inference cost of the KDE estimator to compute the log-likelihood ratio in Algorithm 1 (Line 3) is linear in the number of output samples $m$, as we compute log-likelihood ratio over $2m + 1$ intervals $(\tilde{o}\_i, \tilde{o}\_{i+1})\_{i=0}^{2m}$. This computation cost is significantly smaller than the computation cost for brute-force output set search (that requires exponential in $m$ computation cost to enumerate all possible combinations of intervals $(\tilde{o}\_i, \tilde{o}\_{i+1})_{i=0}^{2m}$).

---

> ### Comment · Reviewer_4UzV · 2024-11-22
> **Comments to authors' responses**
>
> I appreciate the authors' detailed responses.
>
>
> > AR1: Authors' response to my original Comment 1.
>
> **Comment 1:** The Gaussian example cannot be used as the counterexample to conclude that most feasible output set $\mathcal{O}$ cannot be represented by a $\tau'$-log-likelihood-ratio-set.
>
> In fact, for every feasible output set $\mathcal{O}$, there exists a $\tau '\geq 0$ such that
> $$
> \left|\log(\frac{p(x)}{q(x)})\right|\geq \tau ',
> $$
> where
> $$
> \tau' = \inf _{x\in \mathcal{O}} \left|\log(\frac{p(x)}{q(x)})\right|.
> $$
>
> Let $tau'$ be such threhsold associated with $\mathcal{O}$, so that we can denote $\mathcal{O}=\mathcal{O} _{\tau'}$.
> Suppose that $\mathcal{O}$ satisfies $p(\mathcal{O} _\tau) + q(\mathcal{O} _\tau) = p(\mathcal{O}) + q(\mathcal{O})$, **for a given $\tau$**. In addition, let $\hat{\mathcal{O}} = \mathcal{O} _\tau$.
>
> Then, it is clear that we have
> $$
> p(\hat{\mathcal{O}}) + q(\hat{\mathcal{O}}) = p(\mathcal{O} _{\tau'}) + q(\mathcal{O} _{\tau'}).
> $$
> Then, it is easy to verify that Equations/expressions (28), (29), and (30) in Appendix B.3 also holds for $\tau'$:
>
> $$
> \left(1 _{x \in \mathcal{O} _{\tau'} } - 1 _{x \in \hat{\mathcal{O}}}\right) \cdot
> \left( \max\{p(x), q(x)\} - \frac{e^{\tau'}}{1 + e^{\tau'}} \cdot \big(p(x) + q(x)\big) \right) \geq 0
> $$
>
>
> $$
> \begin{aligned}
> &\int _{x \in \mathcal{O} _{\tau'}} \max\{p(x), q(x)\} dx - \frac{e^{\tau'} }{1 + e^{\tau'} } \cdot \big(p(\mathcal{O} _{\tau'}) + q(\mathcal{O} _{\tau'})\big) \\
> &\geq \int _{x \in \hat{\mathcal{O}}} \max\{p(x), q(x)\} dx - \frac{e^{\tau'} }{1 + e^{\tau'} } \cdot \big(p(\hat{\mathcal{O}}) + q(\hat{\mathcal{O}})\big).
> \end{aligned}
> $$
>
> Thus, we have
> $$
> \int _{x \in \mathcal{O} _{\tau'}} \max \{p(x), q(x)\} dx \geq \int  _{x \in \hat{\mathcal{O}}} \max  \{p(x), q(x)\} dx,
> $$
>
> which gives
> $$
> \int_{x \in \mathcal{O} } \max\{p(x), q(x)\} dx \geq \int_{x \in \mathcal{O} _{\tau} } \max\{p(x), q(x)\} dx.
> $$
>
> Thus, for a given $\tau$, we have $\int_{x \in \mathcal{O} } \max\{p(x), q(x)\} dx = \int_{x \in \mathcal{O} _{\tau} } \max\{p(x), q(x)\} dx$.
>
> **Even if** the proof of Theorem 4.3 establishes inequality with both equality and strict inequality, the conclusion applies only to certain specific subset $\mathcal{O}$. There are two key issues:
> 1. First, the existence of $\mathcal{O} _{\tau}$ with $p(\mathcal{O} _\tau) + q(\mathcal{O} _\tau) = p(\mathcal{O}) + q(\mathcal{O})$  for a given $\mathcal{O}$ is not guaranteed.
> 2. Second, for any $\mathcal{O}$ that does have the corresponding $\mathcal{O} _{\tau}$, Equation (6) (assuming it holds with strict inequality as well) only implies that this $\mathcal{O} _{\tau}$ is the optimal output set among all $\mathcal{O}$ that satisfies $p(\mathcal{O} _\tau) + q(\mathcal{O} _\tau) = p(\mathcal{O}) + q(\mathcal{O})$ **for the given $\tau$**. Here, it is the choice $\mathcal{O} _\tau$ that identifies the collection of mathcal{O} that satisfy $p(\mathcal{O} _\tau) + q(\mathcal{O} _\tau) = p(\mathcal{O}) + q(\mathcal{O})$.
>
> Therefore, I must respectfully maintain my original conclusion: The paper does not provide sufficient theoretical claims or methodological support to substantiate the assertion that the proposed approach can identify or approximate the optimal output set.

---

> ### Author Response · Authors · 2024-11-23
> **Response to Theorem 4.3 [existence of optimal output set]**
>
> Thanks for the clarifications, which helped us understand the original comment better -- it establishes a reverse direction equality for Theorem 4.3 -- Eq 6 **at the optimal $\hat{\tau}$**. This is precisely the optimality condition, and does not violate the guarantee that set $\mathcal{O}_{\hat{\tau}}$ dominates any other feasible set $\mathcal{O}$. Below we answer the follow-up comments.
>
>
> > First, the existence of $\mathcal{O}_\tau$ with $p(\mathcal{O}) + q(\mathcal{O}) = p(\mathcal{O}\_\tau) + q(\mathcal{O}\_\tau)$ for a given $\tau$ is not guaranteed.
>
> Existence holds because when $p$ and $q$ are continuous output distributions on $\mathbb{R}$, the following function is continuous with respect to $\tau\geq 0$.
>
> $E(\tau) = p(\mathcal{O}\_\tau) + q(\mathcal{O}\_\tau)$, where $\mathcal{O}\_\tau = \\{x\in\mathbb{R}: \big|\log\frac{p(x)}{q(x)}\big|\geq \tau\\}$
>
> Note that by definition, we have that $E(0) = 2$, $\lim_{\tau\rightarrow+\infty}E(\tau) = 0$, and $p(\mathcal{O}) + q(\mathcal{O}) \in [0,2]$ . Thus by using intermediate value theorem for continuous function, we prove that for any feasible output set $\mathcal{O}$, there exists $\tau\in\mathbb{R}$, such that $p(\mathcal{O}) + q(\mathcal{O}) = p(\mathcal{O}\_\tau) + q(\mathcal{O}\_\tau)$.
>
> We have added this existence statement in Theorem 4.3, and added this proof in Appendix B.2.
>
>
>
> > Second, for any $\mathcal{O}$ that does have the corresponding $\mathcal{O}\_\tau$, Equation (6) (assuming it holds with strict inequality as well) only implies that this $\mathcal{O}\_\tau$ is the optimal output set among all that satisfies $p(\mathcal{O}\_\tau) + q(\mathcal{O}\_\tau) = p(\mathcal{O}) + q(\mathcal{O})$ for the given $\tau$.
>
> This optimality guarantee in Theorem 4.3 indeed is only saying that for any output set $\mathcal{O}$, there exists an $\tau$-log-likelihood-ratio-set that dominates $\mathcal{O}$ in its auditing power (i.e., objective Eq 6). To make this clearer, we have updated the statement of Theorem 4.3 as follows.
>
> Let $p$ and $q$ be two continuous distributions over $\mathbb{R}$. Let $\hat{\varepsilon}(\cdot ; p, q)$ be our auditing objective (4). Given any feasible output set $\mathcal{O}\subseteq\mathbb{R}$, there exists $\tau\in\mathbb{R}$ such that $p(\mathcal{O}\_\tau) + q(\mathcal{O}\_\tau) = p(\mathcal{O}) + q(\mathcal{O})$, where $\mathcal{O}\_\tau$ be the $\tau$-log-likelihood-ratio-set (5) for $p$ and $q$. Further, it satisfies that
>
> $$\hat{\varepsilon}(\mathcal{O}\_\tau; p, q) = \underset{\mathcal{O}\subseteq \mathbb{R}: p(\mathcal{O}) + q(\mathcal{O}) = p(\mathcal{O}\_\tau) + q(\mathcal{O}\_\tau)}{\max} \hat{\varepsilon}(\mathcal{O}; p, q).$$
>
> > Therefore, I must respectfully maintain my original conclusion: The paper does not provide sufficient theoretical claims or methodological support to substantiate the assertion that the proposed approach can identify or approximate the optimal output set.
>
> We believe the reviewer is referring to the observation that Theorem 4.3 only proves that the family of output sets $\\{\mathcal{O}\_\tau\\}_{\tau\geq 0}$ contains the optimal output set $\mathcal{O}\_{\hat{\tau}}$, rather than giving explicit value for the optimal level $\tau^*$. However, we'd like to point out this is already given significant computational benefit for approximating the optimal output set. Specifically, one only needs to search over $\tau$-log-likelihood ratio sets, which is a one-dimensional problem over $\tau\in\mathbb{R}$. This is easier and incurs significantly less computation cost than the original output set optimization problem over all possible output sets $\mathcal{O}\subseteq \mathbb{R}$.

---

> > ### Comment · Reviewer_4UzV · 2024-11-25
> >
> > I thank the authors for their responses.
> >
> > > ... it establishes a **reverse direction** equality for Theorem 4.3 -- Eq 6 at the optimal $\hat{\tau}$. **This is precisely the optimality condition**, and does not violate the guarantee that set $\mathcal{O}_{\hat{\tau}}$ dominates any other feasible set $\mathcal{O}$.
> >
> > Comment: This is incorrect. What I provided above is **not "reverse direction"**.
> >
> > In fact, there are inaccuracies in both (1) the conclusion of Theorem 4.3 and (2) the proof of Theorem 4.3 to establish Eq. (6).
> >
> >
> > (i) Each $\mathcal{O}$ has an **intrinsic threshold** $\tau _{\textup{in}}$, where $$
> > \tau _{\textup{in}} = \inf _{x\in \mathcal{O}} \left|\log(\frac{p(x)}{q(x)})\right|.
> > $$
> >
> >
> >
> > (ii) **Given any $\mathcal{O}$**, define the set: $\mathcal{S}(\mathcal{O})=\{\mathcal{O} _{\tau}, \tau \geq 0| p(\mathcal{O} _\tau) + q(\mathcal{O} _\tau) = p(\mathcal{O}) + q(\mathcal{O}) \}$. That is, $\mathcal{S}(\mathcal{O})$ is the set of all $\tau$-log-likelihood-ratio-sets (with all possible \tau \geq 0) such that  $p(\mathcal{O} _\tau) + q(\mathcal{O} _\tau) = p(\mathcal{O}) + q(\mathcal{O})$ holds for a given $\mathcal{O}$. It is obvious that $\mathcal{O}\in\mathcal{S}(\mathcal{O})$.
> >
> >
> > (iii) Given any log-likelihood-ratio-set $\mathcal{O} _{\tau}$ for some $\tau\geq 0$, define the set $\widehat{\mathcal{S}}(\mathcal{O} _{\tau})=\{\mathcal{O} | p(\mathcal{O} _\tau) + q(\mathcal{O} _\tau) = p(\mathcal{O}) + q(\mathcal{O}) \}$, which is the set of all output sets. It is obvious that $\mathcal{O} _{\tau}\in \widehat{\mathcal{S}}(\mathcal{O} _{\tau})$.
> >
> >
> > First, since every feasible output set $\mathcal{O}$ has $\tau _{\textup{in}}\geq 0$, it is Intrinsically a $\tau _{\textup{in}}$-log-likelihood-ratio-set. Thus, the following claim holds.
> >
> > **Claim 1:** **The existence result or $\mathcal{S}(\mathcal{O})\neq \emptyset$ is trivial** for every $\mathcal{O}$ because $\mathcal{S}(\mathcal{O})$ contains at least $\mathcal{O}=\mathcal{O}_{\tau _{\textup{in}}}$. The existence proof of Theorem 4.3 is not independent of the case when $\mathcal{S}(\mathcal{O})=\{\mathcal{O}\}$.
> >
> >
> > Moreover, {$\mathcal{O} _{\tau}$} _{$\tau \geq 0 $} is a collection of all feasible output sets, because every feasible $\mathcal{O}$ has a corresponding intrinsic $\tau _{\textup{in}}$ such that $\mathcal{O}=\mathcal{O} _{\tau _{\textup{in}}}$. Thus,  {$\mathcal{O} _{\tau}$} _{$\tau \geq 0 $} must contains the optimal output set. Then, the following claim holds.
> >
> > **Claim 2:** Thus, **the statement below Theorem 4.3 "Thus the family of {$\mathcal{O} _{\tau}$} _{$\tau \geq 0 $} contains the optimal output set" is a trivial conclusion that is independent of Theorem 4.3.**
> >
> > Next, suppose that $\mathcal{O}\in \widehat{\mathcal{S}}(\mathcal{O} _{\tau})$ for some $\mathcal{O} _{\tau} \neq \mathcal{O}$. Then, the Eq. (28) also applies as follows:
> >
> > $$
> > \left(1 _{x \in \mathcal{O} } - 1 _{x \in \mathcal{O} _{\tau} }\right) \cdot
> > \left( \max\{p(x), q(x)\} - \frac{e^{\tau _{\textup{in}}}}{1 + e^{\tau _{\textup{in}}}} \cdot \big(p(x) + q(x)\big) \right) \geq 0,
> > $$
> > which is purely from the intrinsic property of each output set and it is **not "reverse direction"**.
> >
> > Following the same steps of Eq. (29)-(30) yields
> > $$
> > \int_{x \in \mathcal{O} } \max\{p(x), q(x)\} dx \geq \int_{x \in \mathcal{O} _{\tau} } \max\{p(x), q(x)\} dx, \forall \mathcal{O}\in \widehat{\mathcal{S}}(\mathcal{O} _{\tau})
> > $$
> >
> > **Claim 3:** Therefore, **Eq. (31) holds only at equality for all $\mathcal{O}\in \widehat{\mathcal{S}}(\mathcal{O} _{\tau})$**.
> >
> >
> > Next, **suppose hypothetically** that the proof of Theorem 4.3 successfully establishes Eq. (6). It is clear that the max on the right-hand side of Eq. (6) is taken over $\widehat{\mathcal{S}}(\mathcal{O} _{\tau})$ for a given $\mathcal{O} _{\tau}$. Then, I have the following claim.
> >
> > **Claim 4:** Eq. (6) states that the log-likelihood-ratio-set $\mathcal{O} _{\tau}$ is (one of) the optimal output set among all output sets in $\widehat{\mathcal{S}}(\mathcal{O} _{\tau})$.
> >
> >
> > Combining **Claim 3** and **Claim 4** gives that what Theorem 4.3 can state is: **given a $\mathcal{O} _{\tau}$**,  **all feasible subsets in $\widehat{\mathcal{S}}(\mathcal{O} _{\tau})$, including $\mathcal{O} _{\tau}$ itself, are equally good.**
> >
> > To sum up, **Theorem 4.3 does not identify the optimal output set for privacy auditing**.

---

> ### Author Response · Authors · 2024-11-25
> **Response to follow up comment**
>
> Thanks for the further clarifications. We saw a few confusions about Definition 4.2 of $\tau$-log-likelihood-ratio-set. We clarify them one-by-one below.
>
>
> > (i) Each $\mathcal{O}$ has an intrinsic threshold $\tau_{in}$, where
> $$
> \tau_{in}=\inf_{x\in\mathcal{O}}\log\left(\frac{p(x)}{q(x)}\right)
> $$
> (ii)  Given any $\mathcal{O}$, define the set: $\mathcal{S}(\mathcal{O}) = \\{\mathcal{O}\_\tau: \tau\geq 0 | p(\mathcal{O}\_\tau) + q(\mathcal{O}\_\tau) = p(\mathcal{O}) + q(\mathcal{O})\\}$. That is, $\mathcal{S}(\mathcal{O})$ is the set of all $\tau$-log-likelihood-ratio-sets (with all possible $\tau\geq 0$) such that $p(\mathcal{O}\_\tau) + q(\mathcal{O}\_\tau) = p(\mathcal{O}) + q(\mathcal{O})$ holds for a given $\mathcal{O}$.
>
> Yes, they are correct. Nevertheless we believe the reviewer meant $\tau_{in}=\inf_{x\in\mathcal{O}} |\log\left(\frac{p(x)}{q(x)}\right) |$, i.e., using **absolute** value (as in Definition 4.2)
>
>
> > (ii, continual)  It is obvious that $\mathcal{O}\in\mathcal{S}(\mathcal{O})$.
>
> This is not true -- output set $\mathcal{O}$ may not follow the structure of $\tau$-log-likelihood-ratio-set (Definition 4.2). For example, under Gaussian distributions $p\sim\mathcal{N}(0, 1)$ and $q\sim\mathcal{N}(1,1)$, by Definition 4.2 (as proved in [our previous response](https://openreview.net/forum?id=A61WjOU7o4&noteId=1qWlTdhQ3g)), for any $\tau\geq 0$, the set $\mathcal{O}\_\tau$ is as follows
> $$
> \mathcal{O}\_\tau = (-\infty, \frac{1}{2} - \tau]\cup [\frac{1}{2} + \tau, + \infty)
> $$
> Given output set $\mathcal{O} = (-\frac{1}{3}, \frac{1}{3})$, we have
> $$
> p(\mathcal{O}) + q(\mathcal{O}) = \Phi(\frac{1}{3}) - \Phi(-\frac{1}{3}) + \Phi(-\frac{2}{3}) - \Phi(-\frac{4}{3}) \approx 0.42147
> $$
> where $\Phi(x) = \underset{Z\sim\mathcal{N}(0,1)}{\Pr}[Z\leq x]$ denotes the CDF of standard normal distribution. The values are computed according to [Gaussian CDF table](https://en.wikipedia.org/wiki/Standard_normal_table).
>
>
> Thus, the set of $\tau$-likelihood-ratio-sets $\mathcal{S}(\mathcal{O})= \\{\mathcal{O}\_\tau: \tau\geq 0 | p(\mathcal{O}\_\tau) + q(\mathcal{O}\_\tau) = p(\mathcal{O}) + q(\mathcal{O})\\}$ (as defined by the reviewer), could be computed as follows.
> $$
> \mathcal{S}(\mathcal{O}) = \Big \\{(-\infty, \frac{1}{2} - \tau^*]\cup [\frac{1}{2} + \tau^*, \infty), \tau\geq 0|1 - \Phi(\frac{1}{2} + \tau) + \Phi(\frac{1}{2} - \tau)  + 1 - \Phi( - \frac{1}{2} + \tau) + \Phi(-\frac{1}{2} - \tau)= 0.42147 \Big\\}= \Big\\{(-\infty, \frac{1}{2} - \tau^*]\cup [\frac{1}{2} + \tau^*, \infty)\Big\\}
> $$
> for a fixed $\tau^*\approx 1.41$. (One can validate via [Gaussian CDF table](https://en.wikipedia.org/wiki/Standard_normal_table) that $1 - \Phi(\frac{1}{2} + \tau^*) + \Phi(\frac{1}{2} - \tau^*)  + 1 - \Phi( - \frac{1}{2} + \tau^*) + \Phi(-\frac{1}{2} - \tau^*) = 1 - 0.97193 + 0.18141 + 1 - 0.81859 + 0.02807  = 0.41896 \approx 0.42147$.
> Observe that $\mathcal{S}(\mathcal{O})$ contains only one set $(-\infty, \frac{1}{2} - \tau^*]\cup [\frac{1}{2} + \tau^*, \infty)$ because the function $1 - \Phi(\frac{1}{2} + \tau) + \Phi(\frac{1}{2} - \tau)  + 1 - \Phi( - \frac{1}{2} + \tau) + \Phi(-\frac{1}{2} - \tau)$ is montonically decreasing with regard to increasing $\tau$. )
>
>
>
> Consequently, $\mathcal{O} \notin \mathcal{S}(\mathcal{O})$ as $(-\frac{1}{3}, \frac{1}{3})\neq (-\infty, -\frac{1}{2} - \tau^*]\cup [\frac{1}{2} + \tau^*, \infty)$.
>
>
>
> > (iii) ...First, since every feasible output set $\mathcal{O}$ has $\tau_{in}\geq 0$, it is Intrinsically a $\tau_in$-log-likelihood-ratio-set.
>
> This is not true. **$\mathcal{O}$ is only a subset of $\tau_{in}$-log-likelihood-ratio-set**, while typically $\mathcal{\mathcal{O}}\neq \mathcal{O}\_{\tau\_{in}}$. E.g., for $\mathcal{O}=(-\frac{1}{3}, \frac{1}{3})$ in [our previous response](https://openreview.net/forum?id=A61WjOU7o4&noteId=1qWlTdhQ3g), one would compute $\tau_{in} = \frac{1}{6}$, and $\mathcal{O}\_{\tau_{in}} = (-\infty, \frac{1}{3}]\cup [\frac{2}{3}, +\infty)$ is a strictly larger set than $\mathcal{O}$. Thus $\mathcal{O}\neq \mathcal{O}\_{\tau\_{in}}$.
>
> > In fact, there are inaccuracies in both (1) the conclusion of Theorem 4.3 and (2) the proof of Theorem 4.3 to establish Eq. (6).
>
> We are happy to further explain any part of the proof or conclusion that the reviewer find inaccurate, if our above clarifications do not address the doubts.
>
> **To sum up, we belive the source of the reviewer's confusion lies in Definition 4.2 for $\tau$-log-likelihood-ratio-set**. We hope we have clarified that
> 1. $\mathcal{O}\_\tau$ is defined on **continuous distributions $p$ and $q$**, thus for each output set $\mathcal{O}$, there **exists** and **only exists one** $\tau$ such that $p(\mathcal{O}\_\tau) + q(\mathcal{O}\_\tau) = p(\mathcal{O}) + q(\mathcal{O})$.
> 2. $\\{\mathcal{O}_\tau: \tau\geq 0\\}$ contains **significantly fewer** sets than all possible output sets $\mathcal{O}\subset \mathbb{R}$. Thus, the **optimality guarantee proved in Theorem 4.3 is non-trivial**.

---

> ### Comment · Reviewer_4UzV · 2024-11-25
>
> Thanks for the detailed responses.
>
> In the Gaussian case where $p \sim \mathcal{N}(0, 1)$ and $q \sim \mathcal{N}(1, 1)$, the log-likelihood ratio is given by:
> $$
> \log\left(\frac{p(x)}{q(x)}\right) = (x - \frac{1}{2}).
> $$
>
> For a threshold $\tau \geq 0$, the log-likelihood-ratio-set $\mathcal{O} _{\tau}$ satisfies:
> $$
> \mathcal{O} _{\tau} = \{x \in \mathbb{R} : |\log(p(x)/q(x))| \geq \tau \}.
> $$
>
> This can be written explicitly as:
> $$
> \mathcal{O} _{\tau} = (-\infty, \frac{1}{2} - \tau] \cup [\frac{1}{2} + \tau, \infty).
> $$
>
> Define the collection $\mathcal{R}$ by $\mathcal{R}$ =  {$\mathcal{O} _{\tau} : \mathcal{O} _{\tau} = (-\infty, \frac{1}{2} - \tau ] \cup [\frac{1}{2} + \tau, \infty), \, \tau \geq 0$}.
>
> **The possible observation that $\mathcal{O} \not\in \mathcal{R}$ does not rule out the fact that $\mathcal{O}$ and its intrinsic threshold $\tau _{\textup{in}}$ satisfy**  {$ x\in \mathcal{O} |  | \log(p(x)/q(x))| \geq \tau _{\textup{in}}$} = $ \mathcal{O}$ or equivalently
> $$
> |\log(p(x)/q(x))| \geq \tau _{\textup{in}}, \forall x\in \mathcal{O}.
> $$
>
> Moreover, **Theorem 4.3 and its proof consider the setting that the choice of $\mathcal{O} _{\tau}$ satisfies**
> $$
> |\log(p(x)/q(x))| \geq \tau , \forall x\in \mathcal{O} _{\tau},
> $$
> which includes any feasible output set $\mathcal{O}$ with its intrinsic threshold. **They are not restricted to the setting that the choice of $\mathcal{O} _{\tau}$ that has specific format such as those in $\mathcal{R}$.**
>
>
> To sum up, my claims in my [previous comment](https://openreview.net/forum?id=A61WjOU7o4&noteId=Z3wxhPGIei) are valid.

---

> ### Author Response · Authors · 2024-11-26
>
> > To sum up, my claims in my previous comment are valid.
>
> The claims only hold for the reviewer's erroneous interpretation of  $\tau$-log-likelihood-ratio set -- we believe the reviewer has confusion between the sufficient and necessary conditions for $\tau$-log-likelihood-ratio set in Definition 4.2. Below we clarify:
>
> >  The possible observation that $\mathcal{O}\notin \mathcal{R}$ does not rule out the fact that $\mathcal{O}$ and its intrinsic threshold $\tau_{in}$  satisfy $\\{x\in\mathcal{O}| |\log(p(x)/q(x))|\geq \tau_{in}\\} = \mathcal{O}$  or equivalently
> $$
> |\log(p(x)/q(x))| \geq \tau_{in}, \forall x\in\mathcal{O}
> $$
>
> This is correct.
>
> > Moreover, Theorem 4.3 and its proof consider the setting that the choice of $\mathcal{O}_\tau$  satisfies
> $$
> |\log(p(x)/q(x))|\geq \tau, \forall x\in \mathcal{O}\_\tau
> $$
> which includes any feasible output set $\mathcal{O}$ with its intrinsic threshold $\tau\_{in}$. They are not restricted to the setting that the choice of $\mathcal{O}\_\tau$ that has specific format such as those in $\mathcal{R}$.
>
>
> This is not true. Theorem 4.3 and its proof considers $\mathcal{O}_\tau$ as constructed in Definition 4.2, which is
>
> $$
> \mathcal{O}\_\tau = \\{x\in\mathbb{R}: |\log\frac{p(x)}{q(x)}|\geq \tau\\}
> $$
> This, by definition, entails two simultaneous requirements for $\mathcal{O}\_\tau$.
> 1. $\forall x\in \mathcal{O}_\tau, |\log\frac{p(x)}{q(x)}|\geq \tau$ (which is what the reviewer wrote).
> 2. $\forall x\notin \mathcal{O}_\tau$, $|\log\frac{p(x)}{q(x)}|< \tau$ (otherwise it contradicts with $x\notin \mathcal{O}\_\tau$). **This requirement is missed by the reviewer**.
>
> In other words, the condition that the reviewer proposed, i.e., $ |\log\frac{p(x)}{q(x)}|\geq \tau$ for any $x\in\mathcal{O}\_\tau$, is only one necessary condition for $\mathcal{O}$ to be a $\tau$-log-likelihood-ratio-set. However, the reviewer has ignored the other necessary condition that $\forall x\notin \mathcal{O}\_\tau$, $|\log\frac{p(x)}{q(x)}|< \tau$ in all their claims 1-4. In fact,  for output set $\mathcal{O}$ with intrinsic threshold $\tau_{in}$, **as long as there $\exists x\notin\mathcal{O}$ such that $|\log\frac{p(x)}{q(x)}|\geq \tau_{in}$**, then $\mathcal{O}$ is not a log-likelihood-ratio-set (per our Definition 4.2).
>
> ---
> Finally, we elaborate in more details that **None of the claims in [the reviewer's previous response](https://openreview.net/forum?id=A61WjOU7o4&noteId=Z3wxhPGIei)  hold for our actual Definition 4.2 of $\tau$-log-likelihood-ratio set.** Specifically, claim 2,3,4 depends on claim 1, and claim 1 is false as we elaborate below.
>
> >  [Claim 1] The existence result or  $\mathcal{S}(\mathcal{O})\neq \emptyset$ is trivial for every $\mathcal{O}$ because $\mathcal{S}(\mathcal{O})$ contains at least $\mathcal{O}$. The existence proof of Theorem 4.3 is not independent of the case when $\mathcal{S}(\mathcal{O})$.
>
> This is false, as we have elaborated [in our last response](https://openreview.net/forum?id=A61WjOU7o4&noteId=q9IUy34C6X) that
> 1.  $\mathcal{O}\notin \mathcal{S}(\mathcal{O})$ (via concrete example)
> 2. By monotonicity, for every $\mathcal{O}$, $\mathcal{S}(\mathcal{O})$ contains **one and one only** output set.

---

> ### Comment · Reviewer_4UzV · 2024-11-26
>
> Thanks for the authors' responses.
>
> The authors are correct that I made a mistake by ignoring the following condition:
> $$
> \forall x\not\in \mathcal{O} _{\tau}, |\log(\frac{p(x)}{q(x)})|<\tau.
> $$
>
> I apologize for the oversights.
>
> Now, it becomes more clear that the proof of Theorem 4.3 indeed proves the Eq. (6) including both equality and strict inequality.
>
> Thus, restricting to the the log-likelihood-ratio sets is without loss of robustness.
>
>
> **Comment 1:** It seems that the choice of log-likelihood-ratio-set and the conclusion that "{$\mathcal{O} _{\tau}$} _$\tau$ contains the optimal output set" aligns with the definition of differential privacy. That is, given $(\epsilon, \delta)$, the entire output set is always partitioned into two subsets. When $\delta=0$, it is clear that the partition of $\mathbb{R}$ by $\epsilon$ includes the "risky" output set satisfying $|\log(p(x)/q(x)|\geq \exp(\tau)$.
>
> It seems that the log-likelihood-ratio-set is a natural choice for threshold-based partitioning of the entire output set \mathbb{R}. However, Section 4.2 still does not identify the optimal output set for auditing. This is because Theorem 4.3 essentially states that the search for the optimal output set should focus on the log-likelihood-ratio-set, which aligns with the fundamental nature of differential privacy.
>
> As also noted by other reviewers, the overall presentation of the paper requires significant improvement. In its current version, the process for identifying optimality is not clearly explained, and Section 4.2 remains confusing. I recommend that the authors revise Section 4.2 to clearly articulate the precise take-home message of Theorem 4.3 and mathematically define the optimality condition for determining the optimal threshold. (Given the approaching deadline, the authors do not need to update their PDF during the rebuttal.)
>
> Although other reviewers have raised additional concerns, since my original review primarily focused on the validity and correctness of Theorem 4.3, I conclude that my original concerns have been sufficiently addressed by the authors. **I will increase my rating.**  Good luck!

---

> > ### Author Response · Authors · 2024-11-27
> >
> > Thanks for acknowledging the clarifications. We are glad the original concerns are addressed. Thanks also for the detailed constructive feedback, we will incorporate them in future revisions.

---

### Official Review · Reviewer_vEYq · 2024-11-03

**Soundness:** 2
**Presentation:** 2
**Contribution:** 3
**Rating:** 6
**Confidence:** 3

**Summary:**

This paper addresses the problem of auditing differential privacy (DP), which involves finding the maximum privacy loss across all possible outputs and possible adjacent datasets. The authors propose a new approach for DP auditing that can be applied with either white-box or black-box access to the mechanism. This approach aims to improve the efficiency and accuracy of DP auditing by optimizing over output sets.

**Strengths:**

The paper identifies an interesting problem in the DP-auditing community. Since the output space can be arbitrarily large, directly finding an output set where the log-ratio is maximized can reduce the number of samples needed.

**Weaknesses:**

**Conceptual confusion:**
- A key weakness of the paper is that it blurs the lines between two distinct concepts: accounting and auditing. For example, Figure 2 wants to provide an improvement over methods that only have black-box access which seems unfair. The authors' method, with access to the distributions, could directly perform accounting and find the exact privacy bound. However, they compare it to black-box access methods that aim to find a lower bound to verify if a mechanism meets a given guarantee. This comparison seems unfair.

- The assumption on access to the densities questions the need for auditing: Having access to the distribution facilitates directly estimating the measure of the set where the density ratio is large, without the need for sampling.

**Misleading claims**:
- Line 69-70 is false. In [4], the authors propose 3 new lower bound methods using only samples.  that do take into account the approximation error. The bounds can be tight for certain distributions. Since a priori the auditor has only black-box access then one cannot relax this.  DP-sniper also incorporates approximation error.
- Not all DP auditing techniques require explicitly finding an optimal output set. Some approaches can indirectly compute privacy bounds using the divergence definition of DP and empirical estimates.

**Limited scope:**
- The paper focuses on DP-SGD, and not more general mechanisms (e.g. exponential mechanisms, histograms, or the sparse vector technique). Their only motivation is computational, but not all mechanisms require training a machine learning model. E.g., reporting the number of COVID cases, counts and aggregates, census statistics, etc.

**Missing references:**
- Introduces an auditing technique based on a regularized renyi divergence.

[1] Domingo-Enrich, C., & Mroueh, Y. (2022). Auditing Differential Privacy in High Dimensions with the Kernel Quantum R\'enyi Divergence. arXiv preprint arXiv:2205.13941.

- Develops upper and lower bounds with white-box access (as this paper assumes).

 [2]Doroshenko, V., Ghazi, B., Kamath, P., Kumar, R., & Manurangsi, P. (2022). Connect the dots: Tighter discrete approximations of privacy loss distributions. arXiv preprint arXiv:2207.04380.

- Develops a statistical test with an approximation error that finds lower bounds on DP parameters:

[3] Property testing for differential privacy
Gilbert, A. C., & McMillan, A. (2018, October). Property testing for differential privacy. In 2018 56th Annual Allerton Conference on Communication, Control, and Computing (Allerton) .

- Introduces three novel tests based on approximations of the renyi, hockey-stick and MMD divergences. All include approximation error:

[4] W. Kong, A. M. Medina, M. Ribero and U. Syed, "DP-Auditorium: A Large-Scale Library for Auditing Differential Privacy," IEEE Symposium on Security and Privacy (SP), 2024.


- What are the propositions related to Yeom et al. (2018); Jayaraman & Evans
(2019); Steinke et al. (2023). And how is Proposition 1 a generalization of this?

**Unclear claims and terminology:**
- The use of the term “MCMC samples” is unclear. Are these simply samples from the mechanism, or is there an underlying Markov chain Monte Carlo method being used? The authors should clarify this terminology.

- The concept introduced in line 51 seems to refer to empirical privacy metrics, which differ from the goal of privacy auditing. Auditing aims to verify whether a mechanism violates its claimed privacy guarantee, rather than assessing the tightness of the bound, as is done in membership inference attacks (MIA) or other inference attacks.


- L.70 the authors say “provides a gain”, a gain in what? Estimation accuracy?

- The second bullet in “Summary of results” seems a direct consequence of the definition of approximate DP and/or DP definition, i.e.,  finding a set where the ratio is large. The same comment applies to Theorem 4.3 seems to be a direct consequence of the definition of DP.


**Questionable Experimental Setup**
- Numerical experiment in 5.2 seems a bad comparison since the authors assume access to the distributions, which allows for direct computation of the exact epsilon, rendering the auditing process unnecessary.
- Experiments have limited scope, testing  only for laplace or gaussian mixtures/distributions and limited comparison to previous work.

- Minor:
  - This sentence could be made clearer:
  - “It ensures that the presence of any individual datum will not be revealed from the probability of any outcome.” A more precise statement is that DP ensures that the presence or absence of any record will not be revealed from the outcome of the mechanism.

- Typos:
  - L145, T: D\in \theta, should be capital \Theta.
  - L 160: “Subselect the scores by a output set”
  - L. 403 “Chi-squared distributions p and q in Appendix D”

**Questions:**

1. Can the authors clarify what they mean by the statement in line 72 that the 'effect of output event set choice on the privacy auditing power is distribution-dependent'? This seems to suggest that the impact of the chosen output set on the effectiveness of the audit depends on the specific mechanism's output distribution. However, in a typical auditing scenario, the auditor does not have a priori knowledge of this distribution. How can one determine the need for finding an optimal output set without such knowledge?

2. What do the authors refer to by MCMC samples? (line 64 and later in the manuscript)

3. Could the authors provide a practical scenario where an auditor has access to the probability densities but would choose to sample from them instead of directly computing the probability ratio for auditing?

4. It is known ([3], [4])  that finding the optimal output set can require exponentially many samples in the worst case. Can the authors elaborate on how their proposed method addresses this potential bottleneck?

5. The authors state in line 77 that 'whether and when optimizing worst-case output sets is elusive.' While this is true for arbitrary distributions, there are cases, such as Gaussian mechanisms, where characterizing these sets is possible. This challenge was a key motivation for developing alternative DP notions like Rényi DP, which provide a smoother measure of privacy and avoid the reliance on worst-case output sets with small measures. Could the authors comment on the connection between their work and these alternative DP notions?

6. Figure 2 suggests that DP-sniper has better outcomes while exhibiting higher variance. In practice you could run several tests and selecting the maximum lower bound could yield better results than the suggested approach (that uses white box information). Could the authors explain the advantages of their approach?

7. Why does Figure 3 compare to DP-Sniper if this method is only for pure DP?

---

> ### Author Response · Authors · 2024-11-22
> **Response to weaknesses [W1-W4]**
>
> Thanks for the valuable feedback.  We provide clarifications to questions below.
> >  [W1] Conceptual confusion:
>
> > A key weakness of the paper is that it blurs the lines between two distinct concepts: accounting and auditing. For example, Figure 2 wants to provide an improvement over methods that only have black-box access which seems unfair. ...
>
> > The assumption on access to the densities questions the need for auditing: Having access to the distribution facilitates directly estimating the measure of the set where the density ratio is large, without the need for sampling.
>
> We think there is a key misunderstanding -- our output set selection Algorithm 1 for privacy auditing **only requires black-box access to Monte Carlo samples** from the output distributions of the DP mechanism. Algorithm 1 first estimates output distribution densities from empirical samples, and then performs output set selection on top of estimated densities. We have updated the pseudocode of Algorithm 1 to make this clearer.
>
>
>
> **Closed-form densities** are used only for theoretically proving the optimality of the log-likelihood-ratio-set (Proposition 4.1 and Theorem 4.3) and for presenting the shape of the theoretical optimal output set (Figure 1). They are not needed for running or evaluating our output set optimization Algorithm 1.
>
>
>
> > [W2] Misleading claims: Line 69-70 is false. In [4], the authors propose 3 new lower bound methods using only samples. that do take into account the approximation error. The bounds can be tight for certain distributions. Since a priori the auditor has only black-box access then one cannot relax this. DP-sniper also incorporates approximation error.
>
> We believe the reviewer is referring to the finite-sample error that is incorporated into the **auditing functions** in prior works via various confidence intervals. This is however, different from incorporating finite-sample error into the **output set optimization objectives**, which none of the prior works ([1,2] as well as DP-Sniper[bichsel2021dp] and [Lu2023general]) achieve to the best of our knowledge. Please see our [response to reviewer 4tEU [W3]](https://openreview.net/forum?id=A61WjOU7o4&noteId=RjcZR8j4kB) for details.
>
>
> > [W3] Limited scope: The paper focuses on DP-SGD, and not more general mechanisms (e.g. exponential mechanisms, histograms, or the sparse vector technique). Their only motivation is computational, but not all mechanisms require training a machine learning model. E.g., reporting the number of COVID cases, counts and aggregates, census statistics, etc.
>
> Due to the significance of DP-SGD in the ML community, we focused on auditing the DP-SGD algorithm and its fundamental building blocks in this paper.  However, we'd like to clarify that our method in principle applies to **any** DP mechanisms, as Algorithm 1 only requires black-box access to Monte Carlo samples from the output distribution.
>
> > [W4] Missing references:
>
> > Introduces an auditing technique based on a regularized renyi divergence.
> [1] Domingo-Enrich, C., & Mroueh, Y. (2022). Auditing Differential Privacy in High Dimensions with the Kernel Quantum R'enyi Divergence. arXiv preprint arXiv:2205.13941.
>
> > Develops upper and lower bounds with white-box access (as this paper assumes).
> [2]Doroshenko, V., Ghazi, B., Kamath, P., Kumar, R., & Manurangsi, P. (2022). Connect the dots: Tighter discrete approximations of privacy loss distributions. arXiv preprint arXiv:2207.04380.
>
> > Develops a statistical test with an approximation error that finds lower bounds on DP parameters:
> [3] Property testing for differential privacy Gilbert, A. C., & McMillan, A. (2018, October). Property testing for differential privacy. In 2018 56th Annual Allerton Conference on Communication, Control, and Computing (Allerton) .
>
> > Introduces three novel tests based on approximations of the renyi, hockey-stick and MMD divergences. All include approximation error:
> [4] W. Kong, A. M. Medina, M. Ribero and U. Syed, "DP-Auditorium: A Large-Scale Library for Auditing Differential Privacy," IEEE Symposium on Security and Privacy (SP), 2024.
>
>
> Thanks for pointing out the references. We'd like to clarify the connections and differences between our work and these related works.
>
> 1. [1,4] are about divergence-based auditing which does not involve any output set optimization. This is orthogonal to the research direction in this paper, i.e., using output set optimization to tighten privacy auditing.
> 2. [2] studies privacy **accounting** rather than privacy auditing, and also assumes **white-box** access to DP mechanism. By contrast, our paper focuses on privacy **auditing** under **black-box** access to the output of the DP mechanism. See our Algorithm 1 pseudocode fo more details. Consequently, the results of [2] are incomparable to ours.
> 3. **We were not aware of the related lower bounds for DP parameters in [3]. We'd appreciate it if the reviewer could give more specific references to the related theorems.**

---

> > ### Comment · Reviewer_vEYq · 2024-12-02
> > **Thanks!**
> >
> > - W1 and W2:  thanks, I think this is clear now.
> >
> > - W3: I agree that DP-SGD is a very important algorithm but I think my point is that for DP-SGD auditing can be done with side information, e.g. knowledge that the noise is Gaussian, and all internals about the algorithm. So comparing with blackbox mechanisms is unfair. I still believe this method can improve over other methods but those were designed to deal with different mechanisms for which we might not have any knowledge about the mechanism. I would suggest either focusing on DP-SGD and then acknowledging this or testing other blackbox mechanisms. I think DP-Sniper and DP-auditorium present several benchmark mechanisms.
> >
> > - W4, [3], see Table 1.

---

> ### Author Response · Authors · 2024-11-22
> **Response to weaknesses [W5-W8]**
>
> > [W4] What are the propositions related to Yeom et al. (2018); Jayaraman & Evans (2019); Steinke et al. (2023). And how is Proposition 1 a generalization of this?
>
> Below we summarize the connections and differences.
> 1. The proof of proposition 4.1. utilizes an inference experiment (Definition B.1) to distinguish between distributions $p$ and $q$. This experiment is similar to [Experiment 1][yeom2018privcay] except for two differences.
>     1. We used two distributions $p$ and $q$ to abstract the distributions for member and non-member scores in the auditing experiment respectively. The randomness of distributions $p$ and $q$ comes from many sources, such as the randomness from the dataset sampling and the output sampling of the DP mechanism. See Table 1 row 1 for examples of i.i.d. samples from $p$ and $q$ in prototypical auditing experiments[jagielski2020auditing,nasr2021adversary].
>     2. We incorporated an output set $\mathcal{O}$ to subselect the output samples for subsequent inference. This changed the guess $\hat{b}_i$ from binary (as in [yeom2018privacy]) to ternary, where the guess $0$ indicates that the output sample $o_i$ is not in the output set $\mathcal{O}$.
> 2. The proof of Proposition 4.1 uses Lemma B.2, which is a generalized variant of prior advantage-based auditing functions in [yeom2018privacy,steinke2023privacy] **from specific designs of output set to any fixed choice of output set**. Specifically, the advantage-based auditing function as defined by [Theorem 1, yeom2018privacy] (and measured by [Section 4, jayaraman2019evaluating]) is equivalent (up to approximation error) to proposition 4.1 under setting the whole output domain as output set, i.e., $\mathcal{O}=\mathbb{R}$. Similarly, under setting $\mathcal{O} = (-\infty, o_{k_-})\cup (o_{k_+}, +\infty)$ where $o_{k_-}$ and $o_{k_+}$ are the bottom-$k_-$ score and top-$k_+$ score in $S_-\cup S_+$, our proposition 4.1 recover the auditing function used under the abstention strategy in [Algorithm 1 and Proposition 5.1, steinke2023privacy].
> 3. Proposition 4.1 proves an auditing function (Eq 3) that explicitly depends on the output set $\mathcal{O}$. By contrast, prior auditing functions (such as [Theorem 5.2][steinke2023privacy] -- also restated in Corollary B.3) only have implicit dependence on the selected output set $\mathcal{O}$. This explicit dependence is the main novelty of Proposition 4.1 which allows us to theoretically analyze the structure of the optimal output set in Section 4.2.
>
>
>
> We have added these discussions in the Appendix (Remark B.4 and Remark B.5) in the revised paper.
>
>
>
> > [W5] The use of the term “MCMC samples” is unclear...
>
> This is a typo and we meant MC (Monte Carlo) samples from the output distributions.
>
> > [W6] The concept introduced in line 51 seems to refer to empirical privacy metrics, which differ from the goal of privacy auditing. Auditing aims to verify whether a mechanism violates its claimed privacy guarantee, rather than assessing the tightness of the bound...
>
> The concept in Line 51 refers to the statistical lower bound $\varepsilon_{LB}$ returned by a privacy auditing experiment, following the notations in [Algorithm 2, jagielski2020auditing]. We agree with the reviewer that this auditing result can examine whether a mechanism violates its claimed DP guarantee. However, in the case that the proved DP guarantee is correct (but not necessarily tight), auditing can also shed light on the tightness of the DP guarantee if the audited lower bound is close to the proved DP guarantee. This is well-discussed in [Section 1.1. The Role of Auditing in DP, jagielski2020auditing].
>
>
> > [W7] L.70 the authors say “provides a gain”, a gain in what? Estimation accuracy?
>
> A gain in terms of higher audited lower bound $\hat{\varepsilon}$ in auditing experiments.
>
>
> > [W8] The second bullet in “Summary of results” seems a direct consequence of the definition of approximate DP and/or DP definition, i.e., finding a set where the ratio is large. The same comment applies to Theorem 4.3 seems to be a direct consequence of the definition of DP.
>
> The objective of output set selection is to enable a higher **audited lower bound**. These auditing lower bounds are **not equivalent to DP definitions**. Instead, auditing functions (such as the ones in Eq 3, Corollary B.3 and Corollary E.2) are interplays between inference performance and finite-sample error on the selected output set $\mathcal{O}$. Consequently, it is not clear as to what output set could maximize the auditing lower bound, e.g., the objective for output set selection in [Eq 4]. This is precisely the question that we analyze in Section 4 of this paper.
>
> We'd also like to refer to our response to [Reviewer 4tEU[W3]](https://openreview.net/forum?id=A61WjOU7o4&noteId=RjcZR8j4kB) for more details on the connections and differences between our work and prior output set optimization objectives.

---

> ### Author Response · Authors · 2024-11-22
> **Response to weaknesses [W9-W11] and Questions [Q1-Q4]**
>
> > [W9] Numerical experiment in 5.2 seems a bad comparison since the authors assume access to the distributions ...
>
> Thanks for pointing out. Our intention was to compare different methods for output set selection at their best performance, i.e. when all methods utilize the density information. However, we agree with the reviewer that for black-box auditing, it is more practical to compare different methods without knowledge about output distribution densities -- **we have updated the comparison in Section 5.2 to compare various methods given black-box samples**. The advantage of our method remains significant, as illustrated by Figure 2.
>
>
> > [W10] Experiments have limited scope ...
>
> We also tested black-box cifar-10 auditing in Section 6, where the output distribution densities are not analytically intractable. We have also compared with all prior works that **perform output set selection** (DP-Sniper[bichsel2021dp], [Lu2023general] and [Steinke2023]). We are happy to compare with any other output set selection method.
>
>
> > [W11] “It ensures that the presence of any individual datum will not be revealed from the probability of any outcome.” A more precise statement is that DP ensures that the presence or absence of any record will not be revealed from the outcome of the mechanism.
>
> We have updated the sentence in the revised paper.
>
>
> > [Q1.1] Can the authors clarify what they mean by the statement in line 72 that the 'effect of output event set choice on the privacy auditing power is distribution-dependent'?
>
> For certain algorithms, such as randomized response, the absolute likelihood ratio magnitude is uniform across the (binary) output domain. Under such mechanisms, one could not obtain higher audited privacy lower bound by optimizing the output set (compared to using the whole output domain). By contrast, for other mechanisms whose absolute log likelihood ratio function is not uniform over the output domain, it is beneficial to select an output set to include regions with higher absolute log-likelihood ratio values, as shown in Figure 1.
>
> > [Q1.2] in a typical auditing scenario, the auditor does not have a priori knowledge of this distribution. How can one determine the need for finding an optimal output set without such knowledge?
>
> When the auditor does not have prior knowledge of the output distribution, we propose to use estimations of the output distribution densities from their empirical Monte Carlo samples for output selection. See Algorithm 1 (Line 1) for details.
>
>
>
> > [Q2] What do the authors refer to by MCMC samples? (line 64 and later in the manuscript)
>
>
> This is a typo and we meant MC (Monte Carlo) samples from the output distributions.
>
> > [Q3] Could the authors provide a practical scenario where an auditor has access to the probability densities but would choose to sample from them instead of directly computing the probability ratio for auditing?
>
> This is not the application scenario of this paper. We focus on privacy **auditing** under **black-box** access to the output of the DP mechanism.
>
> > [Q4] It is known ([3], [4]) that finding the optimal output set can require exponentially many samples in the worst case. Can the authors elaborate on how their proposed method addresses this potential bottleneck?
>
> We are not aware of the lower bounds in [3, 4] about the lower bounds for exponentially many samples in the worst case. We'd appreciate if the reviewer could give more specific references to the related theorems.
>
>
> In terms of the computation cost of our output set selection Algorithm 1, we'd like to comment that Algorithm 1 runs in linear time. Specifically,
>
> - The computation cost for the KDE estimator in Algorithm 1 (Line 1) requires two runs of KDE estimation on $m$ auditing samples, which takes less than a minute on a standard computer when $m$ is in the order $10^6$.
>
> - The inference cost of the KDE estimator to compute the log-likelihood ratio in Algorithm 1 (Line 3) is linear in the number of output samples $m$, as we compute log-likelihood ratio over $2m + 1$ intervals $(\tilde{o}\_i, \tilde{o}\_{i+1})\_{i=0}^{2m}$. This computation cost is significantly smaller than the computation cost for brute-force output set search (that requires exponential in $m$ computation cost to enumerate all possible combinations of intervals $(\tilde{o}\_i, \tilde{o}\_{i+1})\_{i=0}^{2m}$).
>
> We believe that the referred lower bounds [3, 4] would imply that our Algorithm 1 (which is efficient) cannot accurately approximate the output output set in the worst-case. Nevertheless, in our experiments, we observe that Algorithm 1 effectively optimizes the output set for several (possibly non-worst-case) DP mechanisms, including the mixture of Laplace/Gaussian mechanisms (Section 5) and black-box DP-SGD training (Section 6).

---

> > ### Comment · Reviewer_vEYq · 2024-12-02
> >
> > Thanks for the responses. I think the authors improved the presentation of the paper and addressed some of my concerns and confusion. Accordingly, I raised my score. Still, reading other reviewers discussions and the edits, I still think the manuscript and experiments can be presented better to reflect prior work and the contributions of the paper. (Some edits in red have self notes like "add ref" comments).

---

> > > ### Author Response · Authors · 2024-12-03
> > >
> > > Thanks for the concrete pointers and suggestions. We agree that our method's black-box advantage could be highlighted more. We will adjust the presentation order to first discuss black-box auditing of DP-SGD and then use mixture mechanisms as additional examples to show the gain in optimizing output set for black-box privacy auditing.

---

> ### Author Response · Authors · 2024-11-22
> **Response to Questions [Q5-Q7]**
>
> > [Q5] The authors state in line 77 that 'whether and when optimizing worst-case output sets is elusive.' While this is true for arbitrary distributions, there are cases, such as Gaussian mechanisms, where characterizing these sets is possible. This challenge was a key motivation for developing alternative DP notions like Rényi DP, which provide a smoother measure of privacy and avoid the reliance on worst-case output sets with small measures. Could the authors comment on the connection between their work and these alternative DP notions?
>
>
> We completely agree that there exist DP auditing techniques in the literature that do not perform output set selection. However, DP by definition, is a worst-case notion over all output sets. Consequently, to achieve **tight** differential privacy auditing for the worst-case mechanism, it is necessary to perform estimation on an **optimal** output set. For example, divergence is an average notion of information leakage, and it is known that conversion from divergence to DP is loose for the worst-case mechanism [e.g., see Table 1 in [zhu2022](https://proceedings.mlr.press/v151/zhu22c/zhu22c.pdf)]. Thus we do not consider divergence-based auditing in this paper.
>
> The objective of this paper is to investigate the potential of using output set optimization to enable tighter **black-box auditing** for (standard) differential privacy. To this end, the choice of auditing function (whether it is advantage-based or divergence-based) is an orthogonal research direction. Nevertheless, whether divergence-based auditing would benefit from output set selection is an intriguing question. Intuitively, by estimating divergence between conditional distributions (on the selected output set), it may be possible to obtain a tighter DP lower bound (compared to empirically estimated divergences between unconditional output distributions).  We have added this remark in Footnote 1 of the revised paper.
>
> > [Q6] Figure 2 suggests that DP-sniper has better outcomes while exhibiting higher variance. In practice you could run several tests and selecting the maximum lower bound could yield better results than the suggested approach (that uses white box information). Could the authors explain the advantages of their approach?
>
> This is an intriguing question.
> - Firstly, we'd like to clarify that our method (Algorithm 1) only uses black-box information, which is the same as the assumed access by DP-sniper.
> - Secondly, it is worth noting that the **confidence of auditing lower bound would be harmed** by running several tests and selecting the maximum lower bound. For example, let there be $k$ lower bounds that holds with independent probability, where each individual lower bound holds with confidence $1-\beta$. Then the maximum of all lower bounds would only hold with confidence $(1-\beta)^k$. Moreover, if the lower bounds are correlated (e.g., when they are obtained from the same samples), then one can only use the union bound to prove that the maximum of all lower bounds would only hold with confidence $1-\beta k$. In experiments, the sacrifice in confidence could be too high to obtain an improved lower bound estimate, for a fixed desired high confidence level.
> - Nevertheless, we agree that it is an interesting research question as to the potential of tightening privacy auditing via repeated trials for lower bound estimate that has higher variance.
>
>
> > [Q7] Why does Figure 3 compare to DP-Sniper if this method is only for pure DP?
>
>
> - Figure 3 uses our method and DP-sniper to audit privacy lower bound estimates under $\delta=0$. This pure DP auditing setting is precisely where DP-sniper operates (see Section I. INTRODUCTION-Relationship to $\varepsilon$-DP of [Bischsel et al., 2021]).
> - Our Algorithm 1 readily adapts to approximate DP auditing,  as long as the auditing function used in the score set selection step (Line 6 in Algorithm 1) applies to $\delta>0$, i.e., approximate DP. **As an example, we have added the results for auditing approximate DP for the mixture of Gaussian mechanisms in Appendix E.1 of the revised paper.**

---

### Official Review · Reviewer_4tEU · 2024-11-04

**Soundness:** 3
**Presentation:** 1
**Contribution:** 2
**Rating:** 3
**Confidence:** 4

**Summary:**

This paper introduces a framework for improving the accuracy of differential privacy auditing based on trying to identify which canary scores to keep and which to ignore then computing an empirical DP lower bound. Their algorithm efficiently identifies this set when output distributions are known and approximates it from empirical samples when they are not. Experiments on synthetic and real-world datasets, including black-box DP-SGD training, demonstrate that their approach consistently tightens privacy lower bounds compared to existing techniques. The gains are particularly significant with limited auditing samples or when the output distribution is complex (e.g., asymmetric or multimodal).

**Strengths:**

* The problem is interesting and important
* The experimental setups considered cover a good range of problem complexities

**Weaknesses:**

* The manuscript lacks clarity in many places

- The problem statement needs to be more clearly explained, in particular the inner workings of the prototypical auditing experiment. What distribution is the dataset sampled from? How does working with many in- and out-points (usual in MIA) apply to the DP auditing problem where we usually consider two fixed datasets differing in a single point? How does subselection interact with a given auditing function L?

* Focus on pure DP auditing is quite limiting, especially in the context of DP-SGD examples. The manuscript mentions this is for simplicity, but it's unclear from the current presentation whether the methods extend easily or need significant changes.

* Although the manuscript claims "there is no existing consensus regarding the optimal way to incorporate the approximation error with the objective of privacy auditing", there are indeed works that try to incorporate the effect of finite-samples into privacy auditing like [1] and [2]. The manuscript should discuss and compare these methods with the proposed approach.

[1]	William Kong, Andrés Muñoz Medina, Mónica Ribero, Umar Syed: DP-Auditorium: A Large-Scale Library for Auditing Differential Privacy. IEEE SP 2024
[2] NASR, M., SONGI, S., THAKURTA, A., PAPERNOT, N., AND CARLINI, N. Adversary instantiation: Lower bounds for differentially private machine learning. IEEE SP 2021

**Questions:**

* Is the proposed method only applicable to DP mechanisms for ML? Or can it be applied to more general DP mechanisms? (from the presentation it seems to be the former, but it's unclear why)
* What is the Markov Chain in the MCMC sampling refered to in L64-65?
* In Th 4.3: why is there a greater-or-equal rather than equal (since O_tau is feasible)?
* In Fig 1: why is max(p(o),q(o))/(p(o)+q(o)) a relevant quantity to look at?
* In Eq 7: why are the losses l1 and l2 not shuffled using the same permutation as x1 and x2?
* L340: why is a heuristic designed for auditing with a single training run on a large dataset with many "canaries" appropriate to use when auditing with many runs on a 2-point dataset?
* "Let p and q be two probability distributions for member and nonmember scores in the auditing experiment respectively" - what is the randomness over in this distributions? E.g. dataset sampling, mechanism randomness? Without making this more concrete it is not possible to verify the proofs in the supplementary.

---

> ### Author Response · Authors · 2024-11-22
> **Response to weaknesses [W1-W3]**
>
> Thanks for the valuable feedback. Below we answer the questions and clarify our imprecisions.
>
> > [W1] The problem statement needs to be more clearly explained, in particular the inner workings of the prototypical auditing experiment...
>
> Given any fixed auditing experiment, our paper focuses on **optimization of the output set** (Algorithm 1), while keeping the dataset sampling methods and auditing function **unchanged**. See Table 1 for details of how prototypical auditing experiments (Jagielski et al., 2020; Nasr et al., 2021; 2023) can be decomposed into the output set selection component and other components (e.g., dataset sampling methods and auditing function). By modifying the output set selection, we replace the fifth column to Algorithm 1 in Table 1, while keeeping other components the same.
>
> Specifically, Table 1's first row covers the typical DP auditing experiment, which considers two fixed datasets differing in one single point under setting $k=1$. Our numerical experiment, Section 5.2, is also under the setting of i.i.d. samples from output distributions on fixed neighboring datasets, i.e., eq (9), (10), (11), and (12).
>
> > How does subselection interact with a given auditing function L?
>
> We have updated Algorithm 1 (Line 6) to more precisely describe the interactions between our output set optimization algorithm and the auditing function -- the auditing function $L$ is used in Algorithm 1 (Line 6) to select a level $\tau$ that induces output set with the highest auditing function value.
>
> > [W2] Focus on pure DP auditing is quite limiting, especially in the context of DP-SGD examples. The manuscript mentions this is for simplicity, but it's unclear from the current presentation whether the methods extend easily or need significant changes.
>
> Our Algorithm 1 readily adapts to approximate DP auditing,  as long as the auditing function used in the score set selection step (Line 6 in Algorithm 1) is valid for $\delta>0$. **As an example, we have added the results for auditing approximate DP for the mixture of Gaussian mechanisms in Appendix E.1 of the revised paper.**
>
> > [W3] ... there are indeed works that try to incorporate the effect of finite-samples into privacy auditing like [1] and [2]. The manuscript should discuss and compare these methods with the proposed approach.
>
> Thanks for pointing out the interesting related work.
> We believe the reviewer is referring to the finite-sample error that is incorporated into the **auditing functions** in prior works via various confidence intervals. This is however, different from incorporating finite-sample error into the **output set optimization objectives**, which none of the prior works ([1,2], [bichsel2021dp] and [Lu2023general]) achieve to the best of our knowledge. Specifically, **prior works either do not consider the problem of output set selection, or neglect the approximation error in their output set optimization objective**.
> - [1] does not optimize the output event set. Instead, they design function-based testers and dataset finders.
> - [2] does not optimize the output event set. Instead, they use a fixed structure of output set $(-\infty, Z)$ constructed by thresholding the MIA scores (see Table 1 Row 1 for more details).
> - DP-sniper optimizes the output set by an objective function [eq (6) and (7), bichsel2021dp] that solely contains a likelihood ratio term that is independent of the number of samples that fall into the output set. To empirically achieve a small finite-sample error, the authors heuristically selected a threshold $c=0.01$ to ensure that the selected output set has an estimated probability larger than $c$.
> - [lu2023general] uses a similar output set optimization objective [Section 4.2, Algorithm 1] to DP-sniper that does not incorporate the finite-sample error. As an experimental remedy, they perform grid search over different thresholds $c$ to improve the auditing performance.
>
>
> Our work differs from them in our output set optimization objective (Eq 4), which explicitly incorporates the approximation error and its dependence on the output set (Eq 4 second term). This analytical objective is the key ingredient that allows us to **analyze the structure of the optimal output set in Section 4.2, under presence of finite-sample approximation error**. By contrast, prior works (DP-Sniper [eq (6) and (7)][bichsel2021dp] and [Section 4.2, Algorithm 1][Lu2023general]) only prove optimality of likelihood-ratio test for an output set selection objective that directly comes from the DP definition (without incorporating the finite-sample error).
>
> We acknowledge that this is a confusion due to our writing, and we have updated the statement in the paper to "there is no existing consensus regarding the optimal way to incorporate the approximation error into the **output set selection objective** for privacy auditing" to clarify this point.

---

> ### Author Response · Authors · 2024-11-22
> **Response to questions [Q1-Q7]**
>
> > [Q1] Is the proposed method only applicable to DP mechanisms for ML? Or can it be applied to more general DP mechanisms? (from the presentation it seems to be the former, but it's unclear why)
>
> Our method is applicable to general DP mechanisms, as Algorithm 1 only requires black-box access to Monte Carlo samples from the the output distribution. This is also illustrated by our experiments for **black-box** last-iterate auditing of DP-SGD training (Section 6).
>
>
> > [Q2] What is the Markov Chain in the MCMC sampling refered to in L64-65?
>
> This is a typo and we meant MC (Monte Carlo) samples from the output distributions.
>
>
> > [Q3] In Th 4.3: why is there a greater-or-equal rather than equal (since O_tau is feasible)?
>
> Indeed, the greater-or-equal could be strengthened to be eqal.
>
>
> > [Q4] In Fig 1: why is max(p(o),q(o))/(p(o)+q(o)) a relevant quantity to look at?
>
> This term represents the maximum advantage (inference success) of any membership inference on output sample $o$ sampled from $\frac{1}{2}p + \frac{1}{2}q$, where $p$ and $q$ stands for member and non-member output distributions respectively. This inference advantage is the first term of our output set optimization objective [Eq 4], and thus we plot it in Figure 1.
>
>
> > [Q5] In Eq 7: why are the losses l1 and l2 not shuffled using the same permutation as x1 and x2?
>
> The losses l1 and l2 represent the learning task, while x1 and x2 represent the data points. For example, in a pretrain-then-finetune experiment, l1 would represent the loss function for pretraining (next-word prediction), while l2 would represent the loss function for task-specific finetuning (e.g., learning reward function) -- the tasks in a learning procedure are often not permuted despite data shuffling. That is, while x1 and x2 represent data records that could simultaneously be useful for both tasks represented by l1 and l2.
>
> > [Q6] L340: why is a heuristic designed for auditing with a single training run on a large dataset with many "canaries" appropriate to use when auditing with many runs on a 2-point dataset?
>
> We believe the reviewer is referring to our experiments for auditing black-box DP-SGD under one run in Section 6. Indeed, under such settings, to ensure that we are optimizing for the valid auditing lower bound,  we need to use an auditing function that is specifically designed for samples in one-run auditing [Steinke et al., 2023, Theorem 5.2] in our output selection algorithm (Line 6). For completeness, we have restated the auditing function used for the one-run auditing experiment in the Appendix -- see Corollary B.3 and Corollary E.2 for details.
>
>
> To clarify further, given any auditing experiment, our paper only modifies the output set selection component while keeping other components (e.g., dataset sampling methods and auditing function) the same. Consequently, the validity of auditing lower bound is not affected, as long as the output sampling method and the auditing function match the original auditing experiment.
>
> We acknowledge that this is a confusion due to our writing, and we have updated the beginning of Section 6 to clarify.
>
>
>
> > [Q7] "Let p and q be two probability distributions for member and nonmember scores in the auditing experiment respectively" - what is the randomness over in this distributions? E.g. dataset sampling, mechanism randomness? Without making this more concrete it is not possible to verify the proofs in the supplementary.
>
> Proposition 4.1 holds generally under any randomness for member and nonmember scores in the auditing experiment, as long as the member and non-member scores are i.i.d. Monte Carlo samples from $p$ and $q$. We acknowledge that our way of referring to $p$ and $q$ as member and non-member score distributions in auditing experiment could be confusing. We intended to use the two distributions $p$ and $q$ to abstract the randomness for sampling the scores in the auditing experiment. The randomness of distributions $p$ and $q$ comes from many sources, such as the randomness from the dataset sampling and the output sampling of the DP mechanism. See Table 1 for examples of i.i.d. samples from $p$ and $q$ in prototypical auditing experiments[jagielski2020auditing,nasr2021adversary].
>
> We have updated the statement of Proposition 4.1 to remove the unclear reference to the auditing experiment.

---

### Meta-Review · Area_Chair_XZ75 · 2024-12-21

**Metareview:**

This submission provides a framework for improving the accuracy of differentially private (DP) auditing. Experiments on real-world and synthetic datasets are provided, showing improvements in different regimens, including when there are limited auditing samples or when the output distribution is complex.

While the paper studies an important problem and the author(s) answered several of the reviewers' questions, the paper would still benefit from a better exposition and a better coverage of prior related work before being ready for publication.

**Additional Comments On Reviewer Discussion:**

The reviewers raised different concerns about the submission including:

1) Lack in clarity
2) The description of prior related work
3) Whether the paper focuses on privacy auditing or accounting
4) Confusion about Theorem 4.3

While the author(s) satisfactorily addressed several of the reviewers' questions including 3) and 4), concerns remained regarding 1) and 2). The paper would benefit from an improved presentation and description of the prior related work.

---

### Decision · Program_Chairs · 2025-01-22

Reject